# Multiple remote-sensing assessment of the catastrophic collapse in Langtang Valley induced by the 2015 Gorkha earthquake

Hiroto Nagai[1], Manabu Watanabe[2], Naoya Tomii[1], Takeo Tadono[1], Shinichi Suzuki[1]

[1]Space Technology Directorate I, Japan Aerospace Exploration Agency, 2-1-1 Sengen, Tsukuba, Ibaraki, 305-8505, Japan
[2]School of Science and Engineering, Tokyo Denki University, Ishizaka, Hatoyama-machi, Hiki-gun, Saitama, 305-0394, Japan

*Correspondence to*: Hiroto Nagai (nagai.hiroto@jaxa.jp)

**Abstract.** The main shock of the 2015 Gorkha Earthquake in Nepal induced numerous avalanches, rockfalls, and landslides in Himalayan mountain regions. A major village in the Langtang Valley was destroyed along with numerous victims by a catastrophic avalanche event, which consisted of snow, ice, rock, and blast wind. Understanding of hazard process mainly depends on limited witness accounts, interviews, and an in situ survey after a monsoon season. To record the immediate situation and to understand the deposition process, we performed an assessment by means of satellite-based observations carried out no later than two weeks after the event. The avalanche-induced sediment deposition was delineated with calculation of decreasing coherence and visual interpretation of amplitude images acquired from the Phased Array-type L-band Synthetic Aperture Radar-2 (PALSAR-2). These outline areas are highly consistent with that delineated from a high-resolution optical image of World View-3 (WV-3). The delineated sediment areas were estimated as 0.63 km$^2$ (PALSAR-2 coherence calculation), 0.73 km$^2$ (PALSAR-2 visual interpretation), and 0.88 km$^2$ (WV-3), respectively. In the WV-3 image, surface features were classified into 10 groups. Our analysis suggests that the avalanche event contained a sequence of (1) a fast splashing body with an air blast, (2) a huge, flowing muddy mass, (3) less mass flowing from another source, (4) a smaller amount of splashing and flowing mass, and (5) splashing mass without flowing on the east and west sides. By means of satellite-derived pre- and post-event digital surface models, differences in the surface altitudes of the collapse events estimated the total volume of the sediments as $5.51 \pm 0.09 \times 10^6$ m$^3$, the largest mass of which are distributed along the river floor and a tributary water stream. These findings contribute to detailed numerical simulation of the avalanche sequences as well as source identification, and furthermore, altitude measurements after ice/snow melting would reveal a contained volume of melting ice and snow.

**Keywords:** ALOS-2, InSAR, WorldView-3, ALOS World 3D, avalanche, Nepal, Himalaya

## 1 Introduction

A great earthquake of 7.8 Mw, namely the 2015 Gorkha Earthquake, occurred in the district of Lamjung, central Nepal on April 25, 2015 (Ge et al., 2015; Parameswaran et al., 2015), causing more than 9,000 deaths and injuring 23,000 people (Roy et al., 2015). Damage in urban areas especially affected stone/brick masonry structures (Goda et al., 2015), whereas numerous landslides were induced in rural/mountain areas (ICIMOD, 2015a; Kargel et al., 2015).

The most catastrophic collapse on the mountainside was reported in the Langtang Valley, located 70 km north of Kathmandu (ICIMOD, 2015b; Kargel et al., 2015). Landslides, avalanches, and a sudden air pressure wave traveled from a south-facing steep slope to the bottom of a U-shaped valley as described in detail in [2.2] below. The fallen materials, i.e., a mass of boulders, snow, and ice, covered the valley bottom, involving almost all the buildings in Langtang Village. At the opposite side of the valley, trees were prostrated and lost their leaves from the sudden air pressure wave. More than 170 villagers were killed (or left missing) in this event.

Damage detection through the synthetic aperture radar (SAR) technique has previously been applied for urban damaged areas (Kobayashi, 2013; Watanabe et al., 2016; Yonezawa and Takeuchi, 2001) and for landslide, rockslide, and avalanche mountain hazards (Joyce et al., 2009; Metternicht et al., 2005; Riedel and Walther, 2008; Singhroy and Molch, 2004; Wiesmann et al., 2001). However, almost no cases were studied for a large-scale mountain hazard containing multiple phenomena. Therefore, we applied SAR damage detection to this event and evaluated its effectiveness. Fragmentary information from witness accounts, interviews, and an in situ survey after a monsoon season gives some clues; however, scientific understanding of the avalanche process remains poor. Detailed interpretation of the sediment deposition by means of immediately observed high-resolution optical satellite imagery coupled with sediment volume estimation would provide comprehensive spatial and volumetric distribution and a temporal sequence of material deposition. Therefore, in this study, we carried out (1) identification with SAR for an urgent response, (2) the mapping and interpretation of the deposition sequence with high-resolution optical imagery, and (3) volume estimation using the difference between pre- and post-event digital surface models (DSMs). These results allowed us to evaluate the application of SAR to a mountain hazard response and to discuss the overall picture of the avalanche.

## 2 Data set and processing

### 2.1 Study site

The catastrophic collapse was caused in the middle of Langtang Valley (28º12'50"N, 85º30'5"E), one of Nepal's national parks (Fig. 1a). The main river in this valley, the Langtang Khola, flows from east to west and joins the main stream, Bhote Kosi (Trisuli Gandaki), at the end of the valley near a village, Syabru Bensi (Ono and Sadakane, 1986) (Fig.1b). The length of the valley is approximately 50 km, and its width ranges from 1 to 2 km. A typical U-shaped valley was formed by glaciation. The Lirung, Khyimjung, Yala, Shalbachum, Langtang, and Langshisa glaciers are located 4100 m above sea level (a.s.l.). In addition, several unnamed glaciers are distributed along both the ridges of the U-shaped valley (Shiraiwa and Watanabe, 1991), where Mt. Langtang Lirung is the highest peak (7239 m a.s.l.).

The Lantang valley consists of the Gosainkund gneiss zone (various gneisses and granitic migmatite) and the Langtang Himal migmatite zone (medium-grained garnet-mica-gneiss of granitic composition and coarse-grained augen-gneiss) (Le Fort, 1975; Shiraiwa and Watanabe, 1991). Six successive glacial stages were recognized from an in situ dating survey on moraine compositions (Shiraiwa and Watanabe 1991; Shiraiwa, 1994). Relatively extensive glaciation in the Langtang Stage (3650–3000 yr BP) is suggested in the late Quaternary. Permafrost is not highly expected in this valley because of the large amount of winter snow, which prevents deep freezing in winter (Shiraiwa, 1994).

The Langtang Valley is a famous trekking course for tourists, and it has been called "one of the most beautiful valleys in the world." The village of Langtang was called "Yul" by the villagers (Ono and Sadakane 1986). The main local occupations are farming and tourism. Many temporary houses called "Kalkha" were built around the village for livestock farming, i.e., for the transhumance of yaks.

### 2.2 Avalanche event

In this catastrophic event, co-seismic snow-and-ice avalanches and rockfalls with concurrent air blasts occurred (Cadwalladr, 2015). This event consisted of multiple phenomena and was described as a "disaster-within-a-disaster" (Kargel et al., 2015). The sediment deposition consists mostly of accumulated snow and, to a lesser extent, of glacier ice (Fujita et al., 2017). Satellite-based thermal infrared observation conducted 5 days after the quake revealed that the deposition had a 10–20 K lower surface temperature than the surrounding terrain (Kargel et al., 2015). The water stream of the Langtang river was blocked once by the deposition but quickly recovered as the ice and snow deposition melted (Kargel et al., 2015). The materials near the river bed had less boulder- and sand-rich deposition, suggesting that they originated from a snow

avalanche (Fujita et al., 2017). From the sediment volume and catchment area on the mountain hill, the original snow depth before the avalanche occurrence was estimated at 1.82 m in the catchment hillslopes (Fujita et al., 2017). A meteorological observation at a neighboring glacier suggested that four major snowfall events had occurred since Oct 2014, and an anomalous large amount of snow was charged before the quake. An interviewee reported that many hanging glaciers were cracked and huge pieces of ice fell, forming a gathering cloud of snow and rocks with an air blast (Cadwalladr, 2015). However, an in situ survey suggested that detached glacier ice was less dominant than snow, represented by clear ice balls observed in the deposition (Fujita et al., 2017). After a subsequent mass movement between May 8 and 10, melting ice and snow decreased the sediment volume by 40% until Oct, 2015 (Fujita et al., 2017).

Multiple landslides were also reported (Cadwalladr, 2015). Ice cliffs, exposure of ice-rich thick layer under a bolder-rich debris layer, are identified near the Langtang river, suggesting different timing of the avalanche and subsequent rockfalls (Fujita et al., 2017). On the opposite side north-facing steep slopes, debris was found in places 200 m above the deposition bottom, which suggested that it had traveled at 63 m s⁻¹ (Kargel et al., 2015). On the other hand, avalanches dragging sand and silt were reported as "black avalanches" (Fujita et al., 2017). Post-event photographs and satellite images suggested that debris originating from rockfalls and landslides were not dominant in the deposition (Fujita et al., 2017; Kargel et al., 2015).

Related articles all reported trees fallen down in a uniform direction on the opposite side north-facing slope (Cadwalladr, 2015; Fujita et al., 2017; ICIMOD, 2015b; Kargel et al., 2015). This was caused by a catastrophic air blast reaching 332 km h⁻¹ that traveled up to the neighboring villages of Singdum and Mundu (Kargel et al., 2015). The change in location of a boulder during the event suggest that it received a blast exceeding 50 m s⁻¹ (Fujita et al., 2017).

In terms of collapse trigger, three separate main sources were suggested around the mountain peaks at 7000 m a.s.l. by snow cover thinning (~20 m) between April 2014 and May 2015 (Lacroix, 2016). Hanging glacier detachment was considered by another study (Fujita et al., 2017). Furthermore, as described above, anomalous winter snow seemed to amplify the sediment mass (Fujita et al., 2017). Topographic comparisons throughout the event revealed that the total mass of sediment deposition was $6.81 \pm 1.54 \times 10^6$ m³ before the second mass movement caused on May 8-10 (Fujita et al., 2017) and $6.95 \times 10^6$ m³ including the second mass deposition (Lacroix 2016)

## 2.3 Synthetic aperture radar imagery

In order to estimate the damage after the earthquake on April 25, the Japan Aerospace Exploration Agency carried out an emergency observation with the Phased Array-type L-band Synthetic Aperture Radar-2 (PALSAR-2) onboard the Advanced Land Observing Satellite-2 (ALOS-2, "DAICHI-2") aiming at central Nepal at 7:02 on April 26 (GMT). This image was taken in left-looking strip-map mode with a 3-m spatial resolution along the descending orbit of Path 55. Visual interpretation of the orthorectified backscatter amplitude image of HH polarization (product level 2.1) was performed. A pre-event PALSAR-2 image taken at 6:13 on December 28, 2014 was used for comparison. This image was taken in right-looking strip-map mode with a 3-m spatial resolution along the descending orbit of Path 48. Visual interpretation of the orthorectified backscatter amplitude image of HH polarization (product level 2.1) was performed simultaneously.

Not only the amplitude imagery but also the phase information emitted and received by the SAR contributes to situational awareness. We performed a coherence calculation using interferometric phase information of SAR, which was explained by (Plank, 2014) in detail. Coherence can be calculated from two SAR images by observing an identical place twice from the same orbit and incidence angle, thereby achieving similar phase and intensity information of the receiving microwave, which is calculated for a pair of SAR images by

$$\gamma = \frac{E\langle c_1 c_2^* \rangle}{\sqrt{E\langle c_1 c_1^* \rangle E\langle c_2 c_2^* \rangle}} \tag{1}$$

where $c_1$ and $c_2$ are the corresponding complex-valued pixels of the two images, $c^*$ is the complex conjugate of c, and $E$ indicates the expected value. The detailed mathematical procedure is described in (Lopez-Martinez and Pottier, 2007; Touzi

et al., 1999). A significant change in surface features between two observations results in lower coherence (in other words, lower similarity). Other noisy influences, including vegetation growth, can be reduced by calculating normalized differences with a coherence calculated from two pre-hazard images. The normalized coherence decrease (NCD) is calculated as

$$\gamma_{diff} = \frac{\gamma_{pre} - \gamma_{int}}{\gamma_{pre} + \gamma_{int}}$$

(2)

where $\gamma_{pre}$ is the coherence value between two images taken before the earthquake (October 4, 2014 and February 21, 2015), and $\gamma_{int}$ is the coherence value between the two images before/after the earthquake (February 21 and May 2, 2015). These data were acquired from the same orbit with a spatial resolution of 10 m. When $\gamma_{int}$ is calculated for images before/adter a hazard, higher-valued pixels of $\gamma_{diff}$ indicate reduction of the similarity, which has a high potential for hazard-induced deformation or destruction. Several previous studies applied this method using L-band SAR for damage detection in urban areas (Kobayashi, 2013; Watanabe et al., 2016; Yonezawa and Takeuchi, 2001), but no such study applied this method to a mountain hazard. Throughout this study, we aim to emphasize the possibility of normalized conference difference by using L-band SAR for damage detection in mountain regions.

Numerous sources of noise are removed by focal statistics. In the NCD raw image, all pixel values are overwritten by the mean values within 15-pixel circles around each pixel (Fig. 2). This filter emphasizes the concentration of high values, whereas the homogeneously scattered high values are de-emphasized. The detailed steps are as follows:

1.    The radius of a window circle is set as 15 pixels.
2.    A mean value of the pixels in a circle is calculated.
3.    The mean value is placed in the center pixel of the circle.
4.    Moving the circle, every pixel on the output image is filled with the mean values in the same way.

**2.4 Pre-event optical imagery and DSM**

Pre-event optical satellite imagery and its three-dimensional view were generated to grasp the previous situation in further detail. Images of the Panchromatic Remote-sensing Instrument for Stereo Mapping (PRISM) and the Advanced Visible and Near-Infrared Radiometer type 2 (AVNIR-2) onboard the Advanced Land Observing Satellite (ALOS) were acquired at 5:02 on October 12, 2008, which were combined into an orthorectified pan-sharpened image. It is a visible color image with a spatial resolution of 2.5 m. A DSM dataset "ALOS World 3D (AW3D)" was used for this study. AW3D was generated from numerous (>3 million scenes) PRISM nadir, forward, and backward images, which were taken throughout the ALOS operation period (2006 to 2011), automatically stacked, and synthesized into a global DSM dataset with a horizontal spacing of 5 m (Tadono et al., 2015). In addition to its finer resolution compared with existing datasets, such as that of the Suttle Radar Topography Mission and the Global Digital Elevation Model by Advanced Spaceborne Thermal Emission and Reflection Radiometer (ASTER GDEM) (larger than 30 m), occasional anomalous values were excluded from the generation process as well as occasional cloud cover filled with other scenes. As a result, an accuracy of 4.10 m root mean square for the vertical component versus globally distributed ground control points (4622 points) was reported (Tadono et al., 2015). The orthorectified pan-sharpened image was overlain on the AW3D DSM with a pixel spacing of 5 m to show a three-dimensional view for interpretation.

**2.5 Post-event optical imagery and DSM**

Post-event optical satellite imagery and DSM were used to observe the damage situation in detail. A DigitalGlobe's satellite, WorldView-3 (WV-3) observed the Langtang Valley on May 8, 2015, with a panchromatic sensor of 0.31 m spatial resolution and a multispectral sensor of 1.24 m spatial resolution to generate a set of pan-sharpened stereo pair imagery (Fig.

3). First, the area of interest (AOI) is defined so that it includes all sediment depositions. The complicated sediment outlines are delineated from the WV-3 near-infrared band, which shows the best clear contrast between the sediment depositions and the surface terrain, by means of a segmentation function of Iterative Self Organizing (ISO) cluster classifier in ArcGIS (Ball and Hall, 1965; Richards, 2012) (Fig. 4a). Other multispectral band images (red, green, and blue) and the panchromatic image are synthesized into a pan-sharpened image (i.e., color imagery with 0.3 m of spatial resolution). Using this image, the sediment depositions are divided into several groups based on visible characteristics of colors (dark or light) and deposition features (splashing, muddy, and flowing) (Figs. 4b-4f). After all the steps these images and delineated polygon layers are orthorectified with 174 tie points onto the ALOS pan-sharpened image taken on October 12, 2008.

Using the set of pan-sharpened stereo pair imagery, on the other hand, a post-event DSM in a pixel spacing of 2 m was produced by NTT DATA as its commercial service. The DSM is generated by a stereo photogrammetric method using two WV-3 images acquired on May 8, 2015 using stereo-area-collect mode (26.2 km swath, 112 km path). Two images that are (1) forward looking with cross-track tilting to the west hand (i.e., average off-nadir angle: 27º, average target azimuth: 245º /scene id: 104001000BA62E00) and (2) backward looking with cross-track tilting to the west hand (i.e., average off-nadir angle: 27º, average target azimuth: 319º /scene id: 104001000B3B2300) were acquired. Spatial resolution after cross-track tilt was 0.38 m, coarsened from 0.31 m because of tilting. DSM generation flow (i.e., stereo matching, RPC ortho-rectification, pixel resampling, and DSM data output) was performed by NTT DATA with the original software, where the geo-referencing process was supported by WV-3 accurate orbit information without any in situ ground control point and a resampled pixel spacing of 2 m. The officially announced specification shows a vertical accuracy of 4 m and a horizontal accuracy of 5 m as root mean square errors. In two sites neighboring the sediment surface, relative calibration/validation of this DSM and the AW3D DSM was performed and summarized in the supplementary material, in which a standard deviation of 1.5 m between the WV-3 and AW3D DSM is reported.

## 3 Results

### 3.1 SAR amplitude imagery

A post-event PALSAR-2 backscatter amplitude image is shown in Fig. 5a, and a pre-event image is shown in Fig. 5b. The brightness of these images corresponds to the amplitude of the microwave signal reflected to the PALSAR-2 antenna. The difference between the two represents a completely modified surface feature. The mass of the sediment is identifiable only in the post-event image within an area of 0.73 km$^2$ (centroid: 28º12'54"N, 85º30'14"E), ranging approximately 1500 m from upstream to downstream of the Langtang Khola and 700 m from the upper to the lower part of the U-shaped valley.

Comparison of the pre-event PALSAR-2 backscatter amplitude image (Fig. 5b) and an ALOS PRISM/AVNIR-2 pan-sharpened image taken on October 12, 2008 (Fig. 5c) showed bright points and a valley-shaped feature corresponds to buildings and water streams, respectively. These features completely disappear in Fig. 5a, which means that they have been filled by the sediment mass after the quake. A three-dimensional view of the ALOS PRISM/AVNIR-2 image overlaid on the AW3D DSM is shown in Fig. 5d. This sediment area is located below the terminus of a glacier on the north-facing slope. Another glacier flows toward the debris-covered area of the former glacier. This geography suggests the possibility that a large volume of materials traveled from these glaciers with an extremely high potential energy.

### 3.2 SAR coherence decrease

NCD is calculated from the PALSAR-2 images (Fig. 6a). NCD values greater than 0.2 are reclassified into multiple colors with 0.05 steps, in which scattered NDC dots are difficult to be identified. After a focal statistics process with the 15-pixel circle, noisy pixels were moderated, and several parts with a high NCD appeared (Fig. 6b). One of the high-value areas (>0.2) corresponded to the sediment outline delineated from the PALSAR-2 amplitude image (p in Fig. 6b). Separated from

a connecting upper outline at the narrowest part (<5-pixel width), this part has an area of 0.63 km$^2$ with a centroid of [28º12'57"N; 85º30'14"E]. Visual interpretation of the amplitude images (Fig. 5) and NCD calculation (Fig. 6) yield a similar result for the collapsed sediments (Fig. 6b). Moreover, neither method is hindered by cloudy weather therefore, both methods have great potential to immediately indicate a catastrophic collapse and contribute to decision-making for such

hazards in the monsoon season.

Furthermore, above the sediment mass, two areas on the south-facing slope show high-NCD concentrations in (q) 0.22 km$^2$ and (r) 0.07 km$^2$ areas (Fig. 6b). They are located at the downstream periphery of the glacier termini, suggesting a consequent collapse from the tributary glacier to the main glacier. High NCD does not appear on the glacier surfaces, possibly because frequent avalanches and glacier flows cause regular changes in the surface. For such surfaces, the changes

uniquely caused by an earthquake could be identified by NCD calculation.

### 3.3 Collapse mapping with post-event optical imagery

Visual identification and mapping of the sediment depositions from the very-high-resolution WV-3 image resulted in a 0.88-km$^2$ covering, which was classified into 10 groups (A-J) (Fig. 3; Table 1). Group (A) (area: 0.16 km$^2$) is characterized by a dark muddy bottom and splashing uphill parts (Fig. 4b) where numerous trees fallen to the splashing direction are identified

as reported in previous studies (Fig. 4c) (Kargel et al., 2015). Group (B) (area: 0.25 km$^2$) begins from the headwall just under a glacier with a relatively lighter color than (A) (Figs. 4b; 4d). It flows to the river floor with curved streaks (Figs. 3; 4d; 4e). In the river flow, it shows a more mud-like feature with visible wrinkles as group (C) (area: 0.13 km$^2$), accumulating downstream and slightly upstream, maintaining the same color (Fig. 4b). Group (D) (area: 0.14 km$^2$) has a clearly darker surface than (B) and (C) with fewer streaks and several splashed patches (Fig. 4e). A simultaneous, gradual color transition

is seen from (D) to (B) (Fig. 4e). Group (E) (area: 0.02 km$^2$) is located at the lower side of (D) with the same color and a rather muddy feature quite like (C) (Fig. 4e). A gradual color transition is also seen from (D) to (B). On the east side, very darkly colored patches (F) (area: 0.02 km$^2$) and detached parts (G) (area: 0.01 km$^2$) are found (Fig. 3). They seem splashing but have a relatively muddy feature and are not so directionally homogeneous compared with (A). Dark aperture deposition in (H) (area: 0.07 km$^2$) begins from another headwall, which is wider and independent from that of (B) (Fig. 4f). The

splashing parts are blocked by (B) and (C), whereas the western part starts flowing along a narrow path to the river floor grouped as (I) (area: 0.02 km$^2$) (Fig. 3). This flow is finally connected to and covers (C) (Fig. 4c). Group (J) (area: 0.05 km$^2$) is a parallel deposition to (H) and more aperture/splashing compared with (H) (Fig. 4f). The surface color varies from lighter to darker than (J), not related to the flow path.

### 3.4 Surface elevation changes

According to the relative calibration/validation of the AW3D and WV-3 DSMs (see the supplementary material), A horizontal offset of 5 m (1 pixel) to the east and an altitude offset of −27.0 m were added to the WV-3 DSM to achieve the greatest consistency with the AW3D DSM. A possible altitude error of 1.5 m was given to calculate uncertain volumes. After the calibration, the difference between the post-event WV-3 DSM and the pre-event AW3D DSM was calculated and illustrated (Fig. 7a; Table 1). An example of a vertical profile shows sediment deposition on the entire surface terrain from

the foot of the head wall to the river floor, followed by an altitude decrease on the steep, vegetated slope (Fig. 7b), whereas another profile shows the main deposition mainly on the river floor (Fig. 7c).

The spatial distribution of altitude change classified into the 10 sediment groups revealed an increase in surface altitude in the collapse site reaching a maximum of 46.4 m in group (C) (Table 1). The increase in altitude was especially pronounced in groups (C) (mean: 17.8 m) and (E) (mean: 19.7 m), which are muddy depositions located along the river bed. Decreasing

altitude was denoted in groups (A), (F), and (G), where surface erosion and DSM error are dominant. Mean altitude changes in groups (D), (H), and (J) are smaller than the defined uncertainty level of 1.5 m.

Calculating the altitude change and surface area, a total deposition volume of $5.51 \pm 0.09 \times 10^6$ m$^3$ was estimated, which is within the volume range estimated by (Fujita et al., 2017) ($6.81 \pm 1.54 \times 10^6$ m$^3$) and not larger than the volume including the second mass movement ($6.95 \times 10^6$ m$^3$) (Lacroix, 2016). In addition, a total eroded volume of $1.64 \pm 0.06 \times 10^6$ m$^3$ was estimated, most of which belonged to group (A). In addition to the effect of fallen trees, fundamental bias error induced

by WV-3 DSM generation is considered for this extremely steep slope, because splashed patches and muddy deposition both denotes negative values. As well, groups (F) and (G) have a negative net volume difference, possibly because of building collapse and a slightly negative DSM bias larger than the deposition volume of the darkly colored materials.

## 4 Discussion

### 4.1 Temporal sequence of the avalanche event

Identification of sediment deposition layers from the interpretation of a high-resolution WV-3 image suggests that different sources provided various types of deposition continuously in a short period (Fig. 3). The splashing feature of group (A) denotes a uniform scattering direction along lines (x) to (y), suggesting an origin around the cross point of the lines (Fig. 4a). Groups (B) and (C) have a color similar to but lighter than that of (A) and a wider coverage range along line (z) without splashing (Fig. 4a). Thus, a border between (A) and (C) is visually identifiable (p1 in Fig. 4b). The huge mass of (B) and (C)

denoted in Fig. 6a implies slower continuous flow, whereas the negative altitude change on (A) implies fast scattering with an air blast which mowed trees down with less mass deposited on the steep slope. From these we conclude that group (B) was provided after deposition of (A) with a slower speed and larger volume from a different source, which was terminated filling the riverbed in (C) at the end.

Close up, the west side of group (D) has a similar surface feature and color to (B) (p2 in Fig. 4e). In addition, group (E) has a

similar surface feature to that of (C) with some wrinkles having a different surface color. These suggest that the layers of (D) and (E) are much thinner than those of (B) and (C), and they had been carried over the previously deposited (B) and (C).

Group (F) is connected to (D) and a similar dark color to the detached group of (G) (Fig. 3). The group (F) seem to have a smaller amount of mass deposition than (D) at the same or a later time from different sources. There have many ununiform apertures, implying a vertical drop from the source (possibly after hitting some headwall surfaces), rather than a fast second

scattering after first hitting the ground terrain as seen in (A).

Group (H) is distinguished from (B) by its darker color (Fig. 4d) and the beginning headwall foot on the west side (Figs. 3; 4f). Group (I) has a narrow flow with a certain thickness (~10 m) (Table 1) originating from the westernmost part of (H). It is terminated at the river floor, where it pushes and displaces the snout of (C) (p3 in Fig. 4c). These features suggest that (H) and (I) accumulated later than (B) and (C) from another source.

Group (J) has many apertures with a slight flow (fig. 4f). Its lighter and darker colors than (H) are not related to the flow direction, which implies that a heterogeneous mixture of materials was supplied, possibly hitting and involving several origins along the headwall.

Consequently, these considerations give the perspective of a temporal sequence in which the avalanche event provided multiple types of depositions in the following order:

(A) with extremely fast air blast,

(B) with a less flowing speed which covers the entire surface from the hoot of the headwall to the river floor,

(C) as a stacking part of (B) with the least flowing speed along the river flow downstream and partially upstream, depositing a huge mass,

(D) which covers the eastern part of (B) after its deposition with some splashing,

(E) as a terminal part of (D), which covers the eastern part of (C) with a thin layer,

(F) and (G) related to (D), with which splashing in a larger area implies dropping from a relatively higher position,

(G) with which a splashing feature implies the experience of hitting the headwall before deposition,

(H) as the terminal part of (H), riding the terminus of (C) with muddy flowing.

Group (J) is an independent deposition from neighboring (H); however, the deposition timing in the above sequence is unknown. In addition, group (H) has a possibility of earlier deposition than (D) to (G), because groups (D) and (H) have no evident direct relationships.

Thus, very-high-resolution imagery of WV-3 enables reconstruction of the deposition sequence by considering surface feature differences. Although this method, depending on visual interpretation, might contain small differences among individual interpreters, especially on the ambiguous border delineation, the suggested sequence is applied on the vertical profiles of Figs. 6b and 6c, schematically illustrated in Figs. 6d and 6e. Multiple types of avalanche-induced sediments are deposited in layers. The initial sediment, (A) will exist under the following sediment of (B) to (C) along A-A', and additionally (D) to (E) along B-B'. Because remote-sensing techniques have difficulties observing the internal components of sediment, an in situ survey with a boring core and/or ground penetration radar is a possible way to provide some supporting findings. Realistic numerical simulation of avalanche collapse and analysis of heat balance related to the melting process would require consideration of multiple layers precisely mapped by our study. The muddy features interpreted in some layers imply high ice and snow content, which are confirmed in an in situ survey after the monsoon season (Fig. 9). Coupling further altitude measurements with temporal intervals would clarify the surface lowering by melting ice/snow, for which water content estimation is invaluable as part of the input data for avalanche simulation and source consideration.

## 4.2 SAR-derived surface changes

The sediment outline delineated from the PALSAR-2 amplitude image has an area of 0.73 km$^2$, which is very close (i.e., contains an area difference smaller than 5%) to what (Kargel et al., 2015) measured, which slightly underestimates what the WV-3 high-resolution image suggests (0.88 km$^2$). The SAR-derived hazard scale is thus able to be known similarly to those from the optical sensors. The outline delineated from the PALSAR-2 image (0.63 km$^2$) reaches 86% of the area obtained from the WV-3 image (Fig. 8). Its spatial coverage corresponds to groups (B) to (F) and parts of (G) and (J). Most of the flowing parts are included in that coverage, whereas splashed parts beside the central body are excluded (Fig. 8). This suggests that splashed materials are difficult to recognize with the 3-m spatial resolution of PALSAR-2 imagery. Furthermore, group (A) on the north-facing slopes is also ignored. Possibly, microwave reflection from/to PALSAR-2 was hindered by the very steep mountain hillslope.

The sediment outline extracted by NCD calculation has an area of 0.63 km$^2$, which is 72% of that derived from the WV-3 image. Its spatial coverage corresponds to groups (B), (D), (F), (G), and small parts of (A) and (H). NCD indication on the coverage between (F) and (G) can be explained if the surface materials before the avalanche were extensively blown away (e.g., from vegetation or buildings to bare terrain). No NCD indication along the riverbed is explainable by the geomorphic alternation constitutively caused by river erosion, which de-emphasizes the change in the NCD index cause by the avalanche. North-facing slopes were covered by forestry, where little NCD is explainable owing to growing trees and seasonal defoliation. Two extra parts of high-NCD value caused on the valley bottom (i.e., villages of Chyamki and Singdum) ((s) in Fig. 6b) correspond to other collapse occurrences, which were identifiable with an optical image shown in Fig. S6c of (Kargel et al., 2015).

The geospatial information derived from PALSAR-2 is thus usable for disaster response with some inconsistencies, which can be explained by the characteristics of microwaves. A post-event backscatter amplitude image is to be interpreted visually with the assistance of other satellite images, if the interferometric SAR technique cannot be used due to lack of pre-event archived data from the same orbit. The visual interpretation needs careful examination by changing the brightness and contrast of the image, because microwave reflection is influenced simultaneously by both the physical properties and the geomorphic shape of the target area. Through this observation, we have recognized that the visual identification of an

unknown hazard using only images is difficult. Therefore, we needed some information about the place and type of hazard being investigated before interpretation so far. Nevertheless, the approximate scale of the collapse was successfully recognized and provided to the related authorities for emergency response.

Accumulation of the archived data more than once from each orbit allows coherence normalization to emphasize the unusual

decrease in coherence caused by a hazard. Furthermore, removing the small patches by means of focal statistics successfully resulted in sediment extraction, which is reasonably consistent with that from a high-resolution satellite image. For other kinds and scales of hazards, the parameter settings (i.e., circle size to obtain a mean value) need to be assessed and validated through several case studies. The NCD calculation cannot be used for slightly but constitutively changing terrains, such as river banks and vegetation.

In terms of the avalanche source, we found two areas with high-NCD values above the collapsed sediment on the hillslope ((q) and (r) in Fig. 6b). On the glacier surfaces, regular flow could de-emphasize the unusual change caused by the avalanche or other collapse events, even if they had happened. The two parts suggest that the falling materials went through these parts and altered their surface features. We could not detect drastic surface changes in the uphill between AW3D (pre-event) and WV-3 (post-event) images, but Lacroix (2016) supported our suggestion by comparison of DSMs generated from Satellite

Pour l' Observation de la Terre (SPOT). On such an extremely steep slope, it is difficult to obtain an accurate DSM from most satellite images; therefore, NCD coupled with DSM comparison would be preferable.

### 4.3 Validation of volume estimation

Kargel et al. (2015) estimated the total mass of this sediment as ~$3.3 \times 10^9$ kg, assuming a homogeneous thickness of 2 m on its entire surface. Our study revealed a heterogeneous volume distribution, which was especially concentrated along the

water streams. According to the assumed density of 2200 kg m$^{-3}$ (Kargel et al., 2015), our estimated volume ($5.51 \times 10^6$ m$^3$) turns out to be $12.1 \times 10^9$ kg, reaching 3.7 times of the former (Kargel's) estimation. Our volume estimation is smaller than that by (Lacroix, 2016) through a comparison of SPOT-derived DSMs ($6.95 \times 10^6$ m$^3$), possibly because the second avalanche was caused between May 8 and 10 (Fujita et al., 2017). Furthermore, (Fujita et al., 2017) performed an in situ survey from on which they estimated the total volume of the first avalanche sediment as $6.81 \pm 1.54 \times 10^6$ m$^3$, with a

possible range covering our estimated value.

The post-hazard WV-3 DSM in our analysis and the SPOT-derived DSM by Lacroix (2016) show the situation very soon after the hazard. This is valuable because following in situ surveys would measure the height after the internal ice/snow deposition started melting. These remote-sensing techniques enable repeat observations of the same accuracy at a lower cost than an in situ survey. Coupling of the regular surface-lowering measurements and assessment of thermal properties of the

sediment would contribute to understanding of the internal composition of ice/snow and rock materials, whether they have similar characteristics with those of a debris-covered glacier, for example. (Fujita et al., 2017) demonstrated that image acquisition from unmanned aerial vehicles resulted in accurate DSM generation compared with in situ dGPS measurement. Such an in situ campaign would validate satellite-derived DSM accuracy.

### 5 Conclusion

Initial multi-satellite observation and assessment were carried out for the catastrophic avalanche induced by the 2015 Gorkha Earthquake in Nepal on April 25. Radar observation by means of PALSAR-2 resulted in two successful methods of backscatter amplitude image and coherence analysis, which are both usable for an urgent hazard response to acquire quantitative information and not hindered by frequent cloud cover in those regions during monsoon season. Detailed visual interpretation and classification of the sediments with a WV-3 pan-sharpened image indicated multiple sediment layers,

which suggest sequential failure of materials from different sources. The difference between the pre- and post-event DSMs

estimated a total sediment volume of $5.51 \times 10^6$ m$^3$ (and a weight of $12.1 \times 10^9$ kg), of which the dominant mass is accumulated along the river streams. A negative altitude change suggests erosion and denudation of surface objects and measurement error in difficult topography for DSM generation. Quantitative spatial and volumetric assessments and classification derived from this study would contribute to further studies such as avalanche simulation, melting process

estimation, etc.

**Author contributions**

H. Nagai designed the total analysis and discussion. M. Watanabe carried out InSAR coherence calculation and improved the manuscript. N. Tomii organized the use of WorldView-3 image. T. Tadono and S. Suzuki managed the ALOS-2 observation.

**Competing interests**

The authors declare that they have no conflict of interest.

**Acknowledgment**

We greatly appreciate the financial support from the Council for Science, Technology, and Innovation, Cross-ministerial Strategic Innovation Promotion Program (SIP), "Enhancement of societal resiliency against natural disasters" (Funding agency: Japan Science and Technology agency). WorldView-3 image was distributed by DigitalGlobe, Inc. WorldView-3

digital surface model was generated by NTTDATA CORPORATION, included DigitalGlobe, Inc. ALOS World 3D dataset was generated by NTTDATA CORPORATION and Remote Sensing Technology Center of Japan included JAXA. We thank Dr. K. Tsutsui, M. Ozaki, and their colleagues at NTT DATA CORPORATION Inc. for providing technical information. We also thank the members of J-RAPID team "Investigation of cryo-geohazards in Langtang Valley, Nepal" founded by the Japan Science and Technology Agency and those of a non-governmental organization, Langtang Plan, for valuable

information from their post-hazard in situ survey. A part of this research is conducted under the agreement of Japan Aerospace Exploration Agency (JAXA) Research Announcement titled "The 6th Research Announcement (RA-6) for the Advanced Land Observing Satellite-2 (ALOS-2)". We thank the editor and anonymous reviewers for their valuable comments and handling of review process.

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

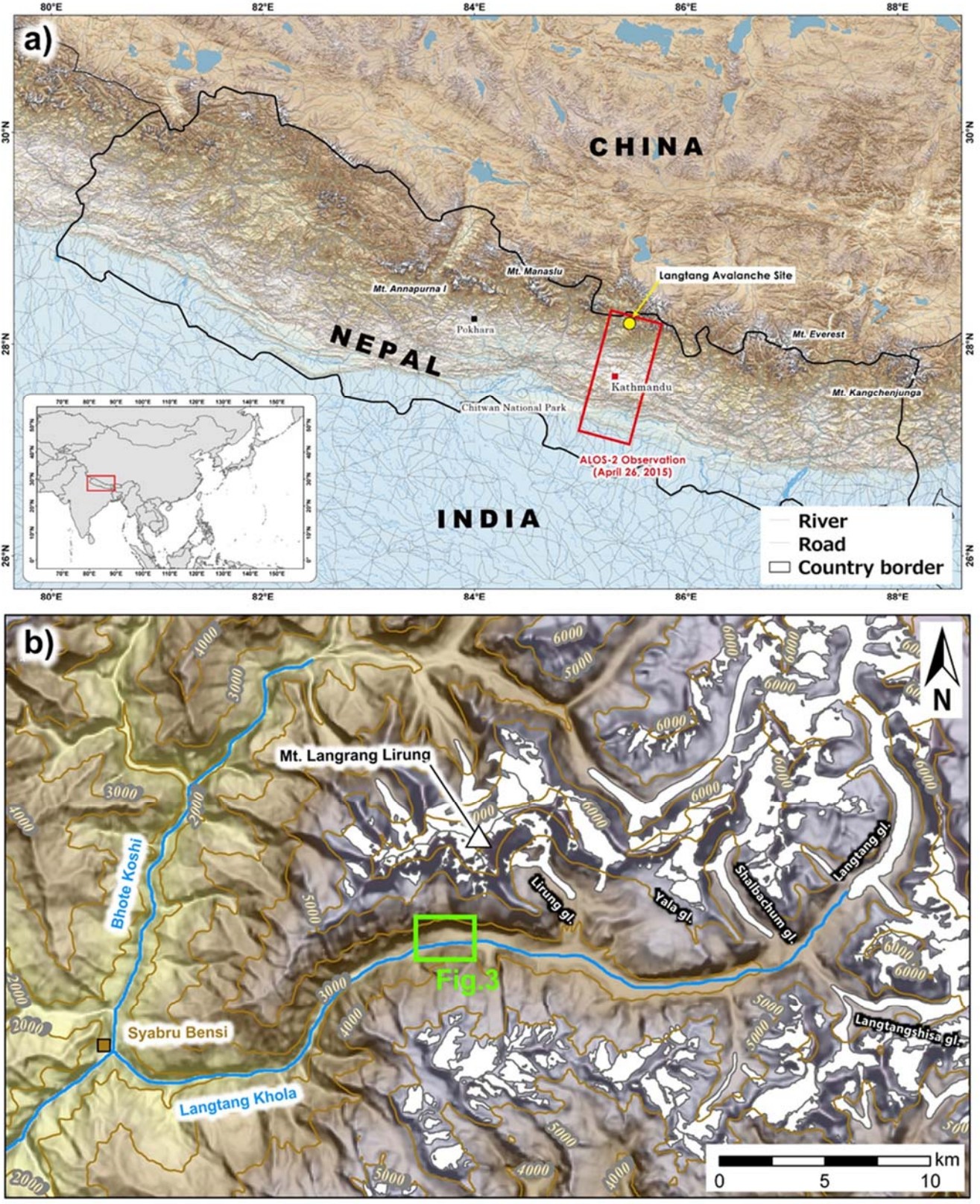

**Figure 1: Location of the study site (a) in the Nepal Himalaya and (b) its close-up view. Topography obtained from NASA/USGS SRTM3v4, glacier outlines obtained from ICIMOD Mountain Geoportal, rivers and roads obtained from DIVA-GIS (partially revised), and country borders obtained from ThematicMapping.org are illustrated.**

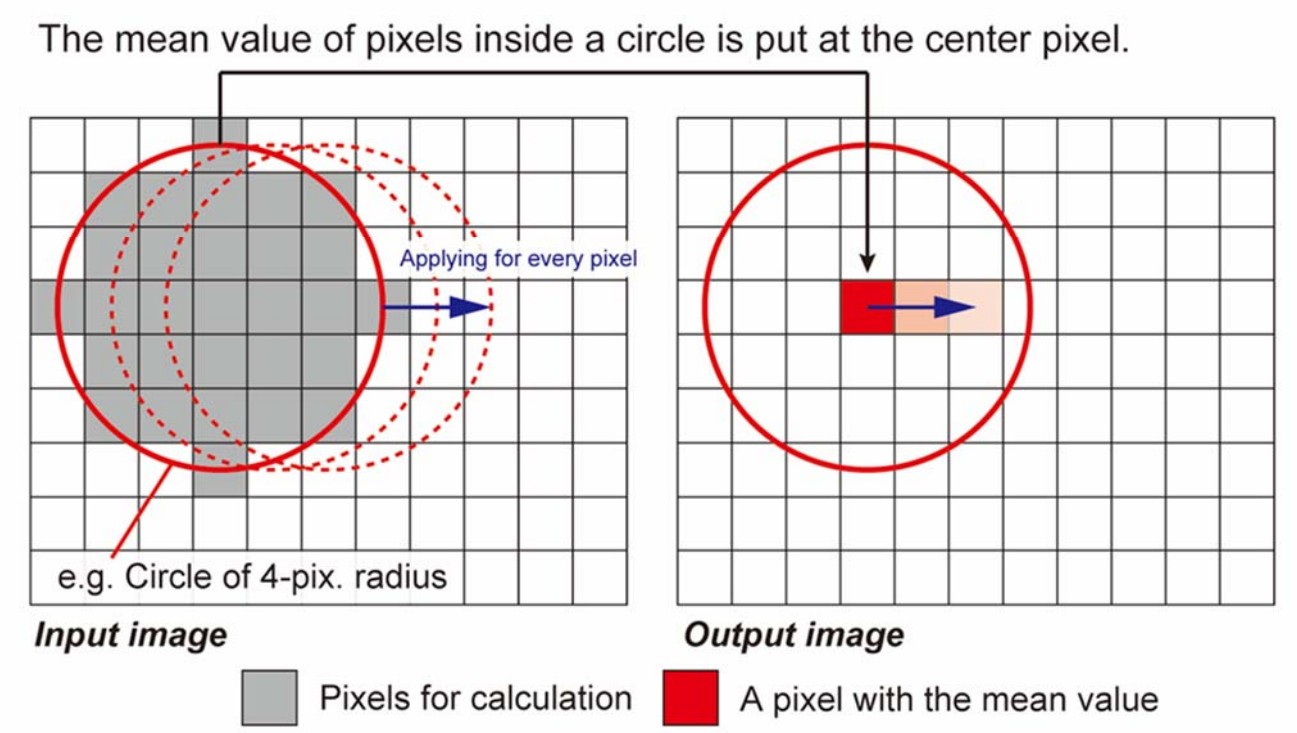

**Figure 2: Concept of a focal statistics filter to reduce noise.**

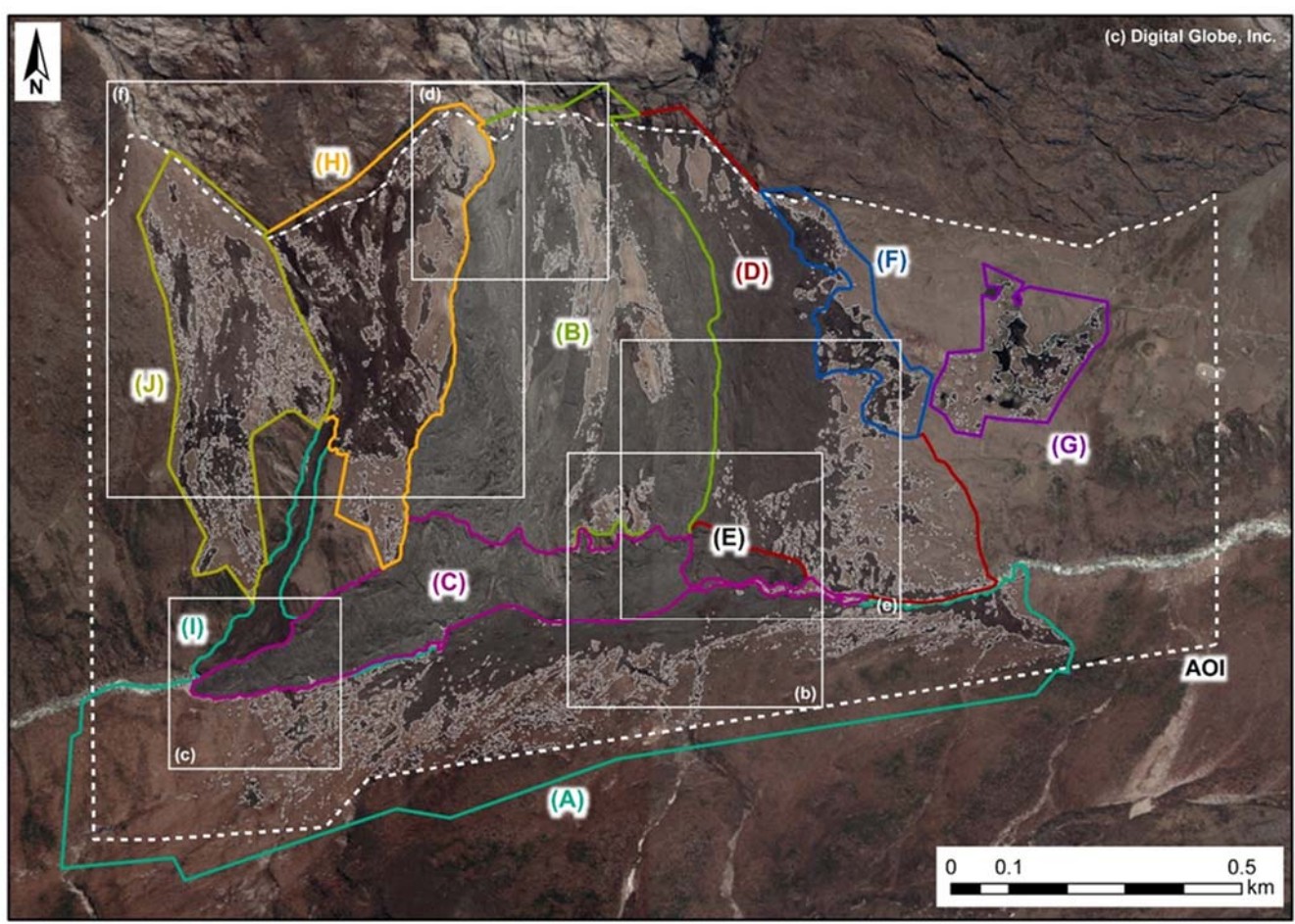

**Figure 3: Avalanche-induced sediment deposition observed with World View-3. The dotted rectangle is the area of interest (AOI)**
5    **of this study, within which sediment depositions are classified from (A) to (J). Squares correspond to the close-up view in Fig. 4.**

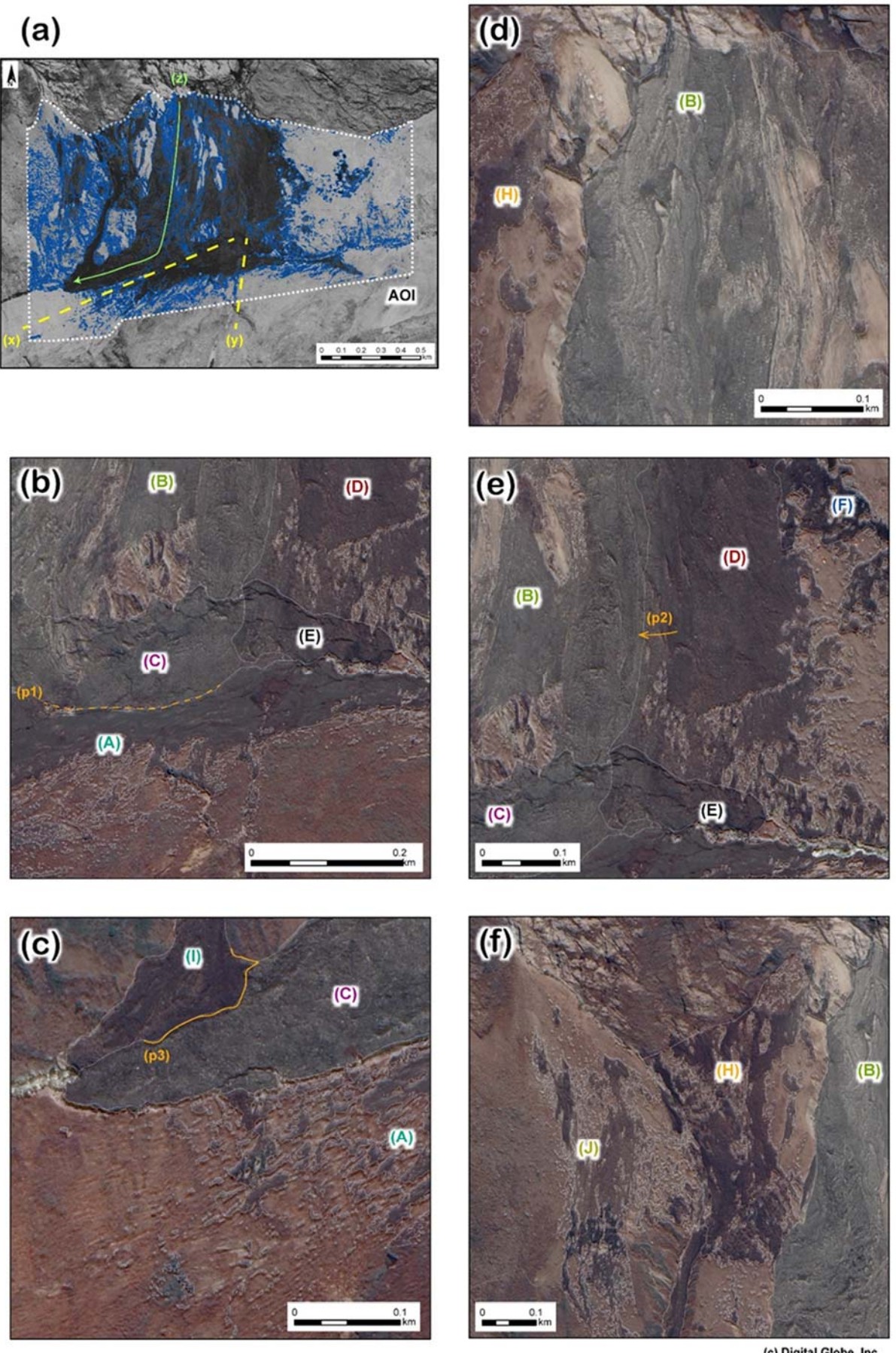

**Figure 4: Details of sediment mapping. (a) Unsupervised classification was initially applied to the WV-3 thermal infrared image. (b-f) Detailed interpretation and classification were carried out by comparing the differences in sediment colors and physical characteristics.**

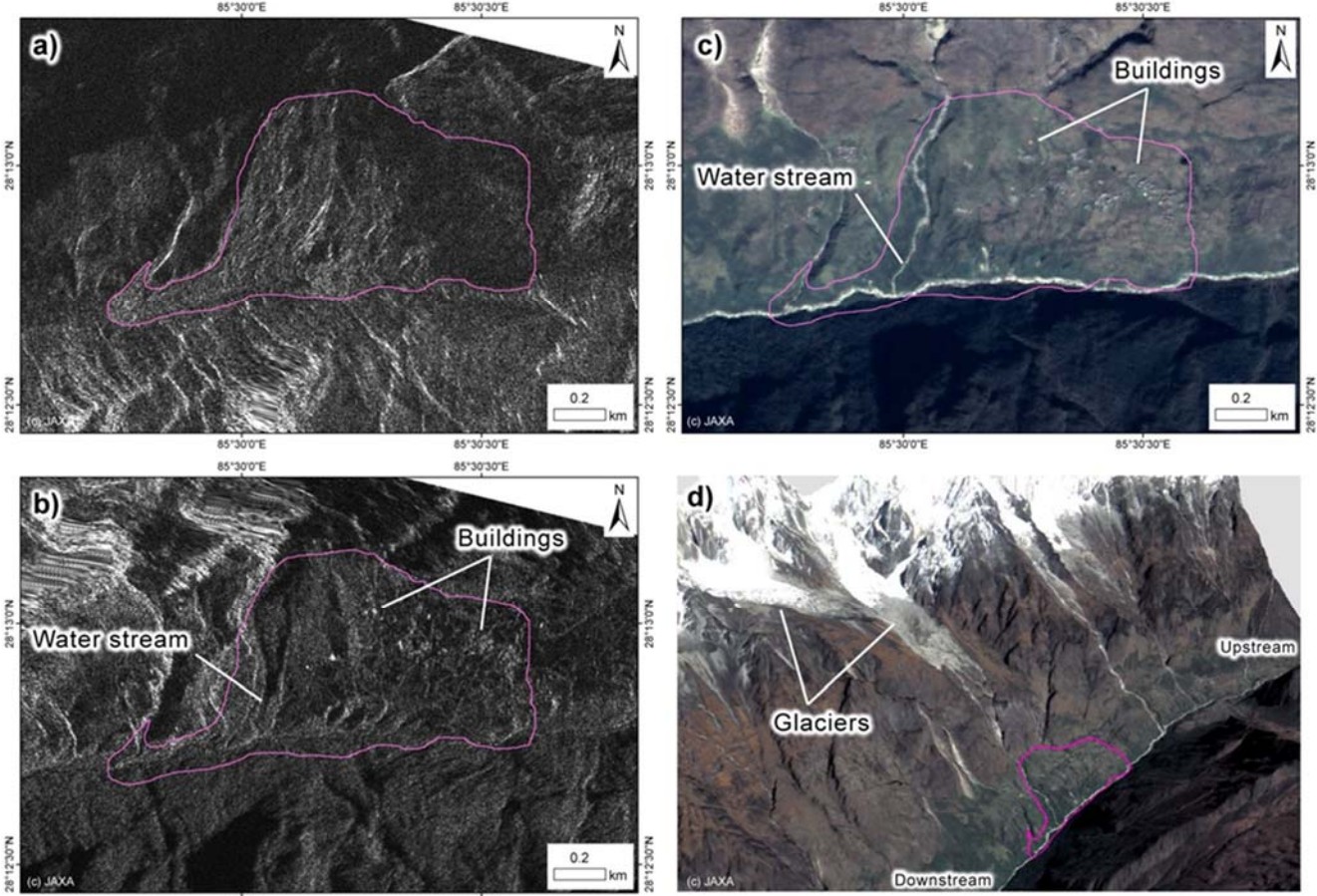

**Figure 5: Collapsed sediment identified on ALOS/ALOS-2 images. Difference between PALSAR-2 images of (a) post-quake (April 26, 2015) and (b) pre-quake (December 28, 2014) denotes that the buildings and water stream identifiable by (c) a pre-quake ALOS pan-sharpened image (October 12, 2008) has been covered by sediments. (d) a three-dimensional view of the ALOS pan-sharpened image overlaid on ALOS World 3D digital surface model.**

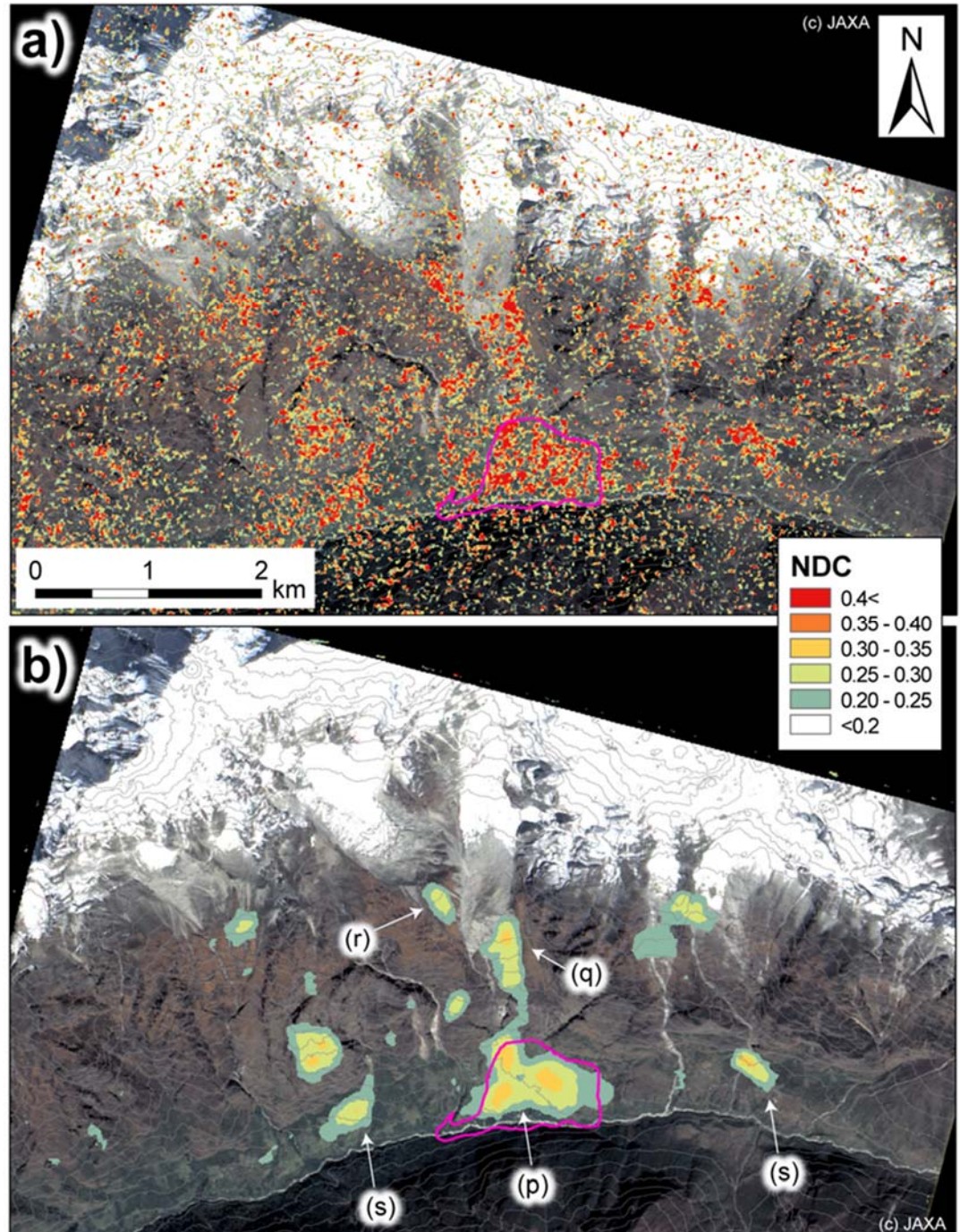

Figure 6: Normalized coherence decrease around the collapsed area in Langtang Valley. (a) Original output image is processed into two images of focal statistics with (b) 15-m circles for calculating the mean values. High-value parts are denoted on (p) the collapse area, (q) a glacier moraine, and (r) that of a tributary glacier, as well as (s) two places in the valley bottom. The background is an ALOS pan-sharpened image observed on October 12, 2008.

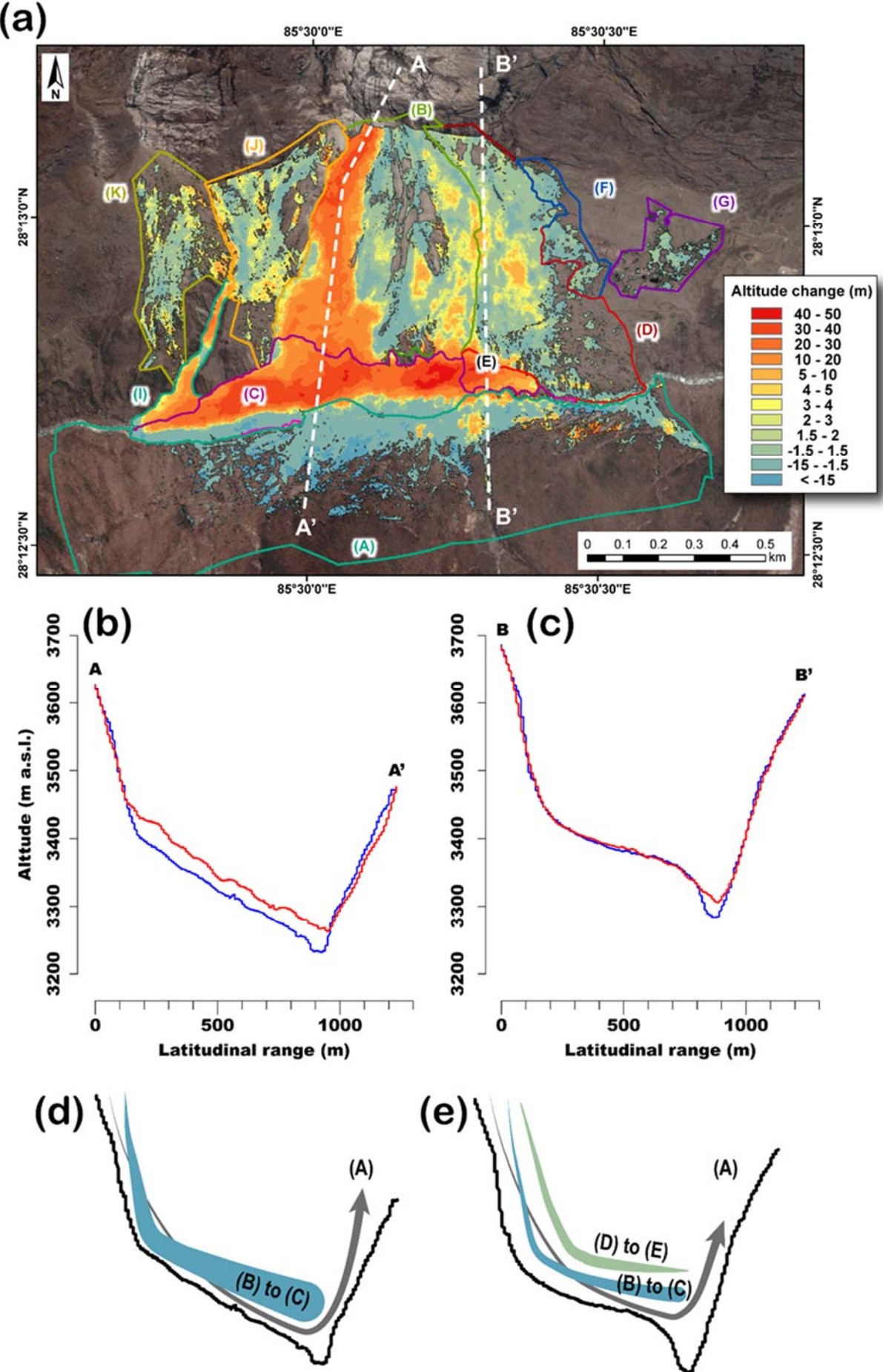

**Figure 7: (a) Estimation of the sediment volume from the altitude difference between pre-event ALOS World 3D DSM and post-event WV-3 DSM. Vertical profiles along (b) A-A' and (c) B-B' are used respectively for (d) (e) schematic illustration of avalanche sequences.**

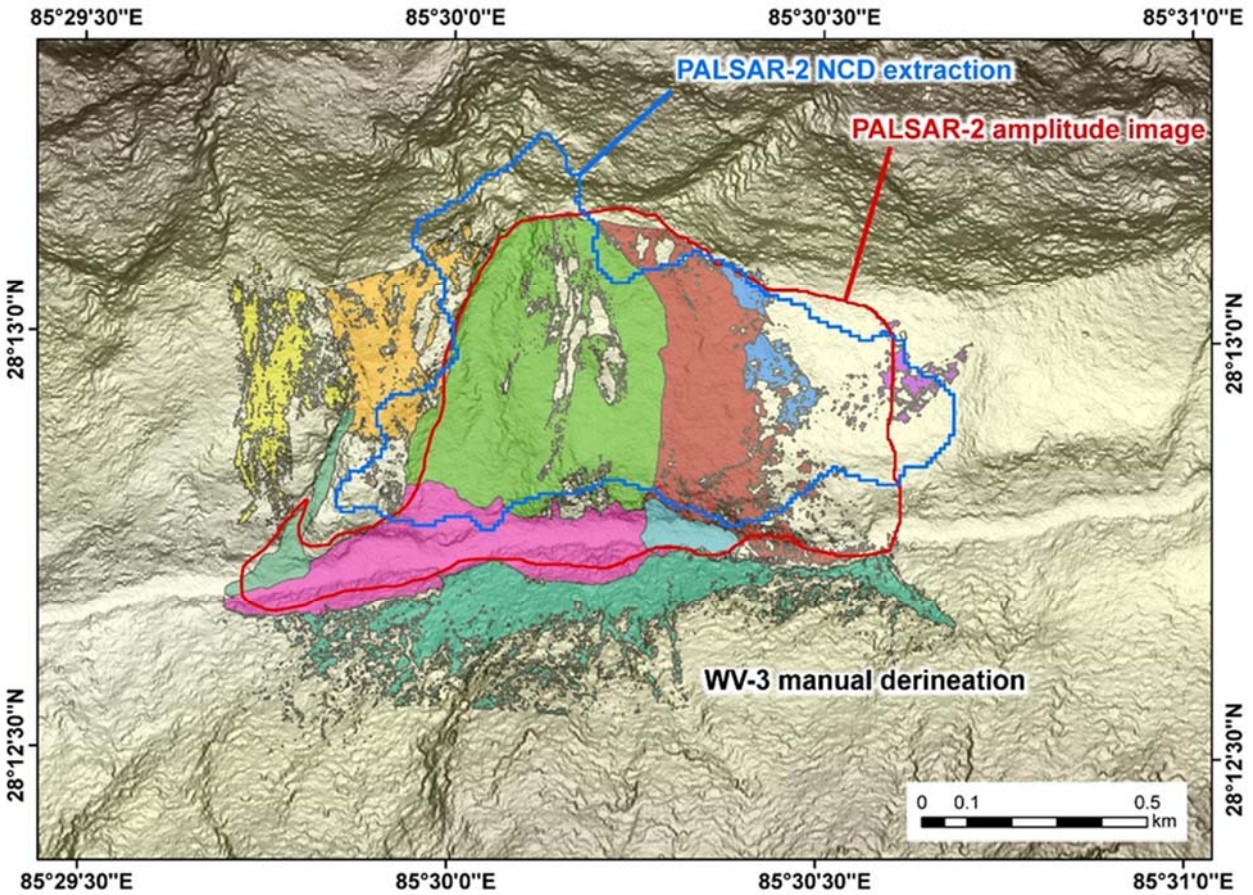

**Figure 8: Sediment outlines delineated from PALSAR-2 data and WorldView-3 (WV-3) imagery. The background is hill-shade imagery generated from the AW3D DSM.**

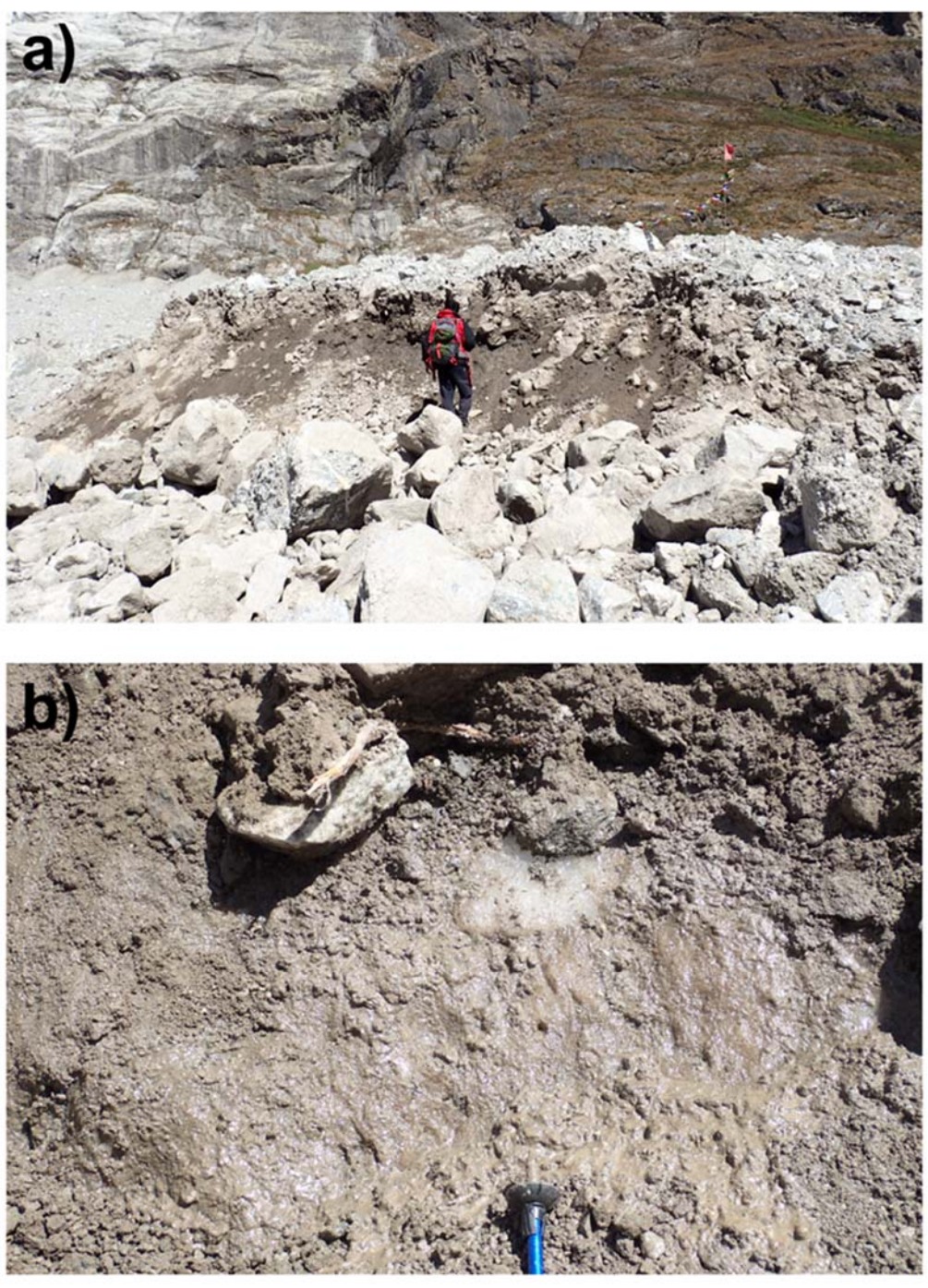

**Figure 9. (a) A image of the avalanche-induced sediment surface taken in an in situ survey carried out for/by Fujita et al. (2017). (b) A close-up image showing exposed and melting ice [Date: Oct. 21, 2015].**

**Table 1: Identifiable surface color and physical features, area, thickness, and volume changes for the 10 classified deposition groups.**

| Group | Colour | Surface feature | Area (km$^2$) | Altitude change (m) Average | Altitude change (m) Mean | Volume change (10$^6$ m$^3$) Deposition | | | Volume change (10$^6$ m$^3$) Erosion | | | Volume change (10$^6$ m$^3$) Net | | |
|---|---|---|---|---|---|---|---|---|---|---|---|---|---|---|
| (A) | Dark | Muddy to splash | 0.16 | 28.1 | -9.3 | 0.09 | ± | 0.02 | 1.14 | ± | 0.15 | -1.05 | ± | 0.15 |
| (B) | Light | Flowing with streaks | 0.25 | 42.9 | 7.5 | 1.99 | ± | 0.23 | 0.11 | ± | 0.05 | 1.88 | ± | 0.24 |
| (C) | Light | Muddy | 0.13 | 46.4 | 17.8 | 2.53 | ± | 0.15 | 0.17 | ± | 0.03 | 2.36 | ± | 0.16 |
| (D) | Dark, gradually light | Less flowing and few streaks | 0.14 | 23.8 | 0.5 | 0.13 | ± | 0.06 | 0.07 | ± | 0.04 | 0.06 | ± | 0.07 |
| (E) | Dark, gradually light | Muddy | 0.02 | 41.7 | 19.7 | 0.30 | ± | 0.02 | 0.00 | ± | 0.00 | 0.30 | ± | 0.02 |
| (F) | Very dark | Muddy and splash | 0.02 | 3.7 | -1.8 | 0.00 | ± | 0.00 | 0.03 | ± | 0.02 | -0.03 | ± | 0.02 |
| (G) | Very dark | Detached, muddy and splash | 0.01 | 1.4 | -1.7 | 0.00 | ± | 0.00 | 0.01 | ± | 0.01 | -0.01 | ± | 0.01 |
| (H) | Dark | Muddy and splash | 0.07 | 39.2 | 1.3 | 0.09 | ± | 0.04 | 0.04 | ± | 0.02 | 0.06 | ± | 0.04 |
| (I) | Dark | Muddy | 0.02 | 30.6 | 10.5 | 0.21 | ± | 0.03 | 0.00 | ± | 0.00 | 0.20 | ± | 0.03 |
| (J) | Light to dark | Muddy and splash | 0.05 | 13.2 | 1.4 | 0.05 | ± | 0.02 | 0.01 | ± | 0.00 | 0.04 | ± | 0.02 |
| Total | | | 0.88 | 46.4 | 4.0 | 5.51 | ± | 0.09 | 1.64 | ± | 0.06 | 3.87 | ± | 0.11 |