# Peer review of "Multiple remote-sensing assessment of the catastrophic collapse in Langtang Valley induced by the 2015 Gorkha earthquake"

_Natural Hazards and Earth System Sciences, 2016_

## Referee Comment (RC1) · Anonymous Referee #1 · 9 Oct 2016

General comments

This paper demonstrated an assessment of the sediments caused by a catastrophic avalanche, using Remote Sensing data, such as, ALOS-2, WorldView-3, ALOS World 3D, etc. The topic of this manuscript is quite interesting, because L-band (PALSAR-2) could penetrate the cloud and vegetation. in fact, catastrophic collapse (earthquake, debris flow, landslide, etc.) always seem to be associated with rain and vegetation. So, PALSAR-2 have a great potential to immediately indicate a catastrophic collapse and contribute to decision-making for such hazards in the monsoon season. However, this manuscript need more information to illustrate its conclusions. Below, I comment on the few things which I think can be improved.

Specific comments

(1) "Introduction", in this section, introduced too many information about study site (move it to the 2.1 section), but lack the background and innovation to this research, it can't attract the reader's interest immediately. (2) "2.1 study site", I think you'd better add a location map of study site to help to understand where is it. (3) "2.2 Synthetic aperture radar imagery", just defined normalized coherence decrease (NCD), didn't explain what is Coherence calculation and how to calculate it, in addition, you can't leave out the process and method to noises filter, it's too brief in this part. (4)" 2.4 Post-event optical imagery and DSM", the post-event DSM is very important to calculate the sediments volume, this paper just said "was produced by NTT DATA as its commercial service", obviously it's not enough, And "relative calibration/validation of this DSM and the AW3D DSM was performed and summarized in a supplementary material", i didn't find the supplementary material. (5) Is it possible to do field survey to verify the results? (6) Improve the quality of the figures

---

## Referee Comment (RC2) · Anonymous Referee #2 · 23 Oct 2016

In this manuscript, the authors describe the use of different remote sensing approaches for the identification of the effects of the 2015 Gorka Earthquake. In my opinion, the topic is very interesting and suitable for this journal, but the manuscript could be considered ready for the publication only after major revisions. In the following some suggestions for the authors: Page 1 line 30: in the abstract the authors describe an avalanche and they introduce that the paper will be focused on it. After, in the introduction, they introduce the presence of avalanche, but also landslides and other gravitational processes. For the reader is not very easy to understand which what happened in this area and then to follow the authors in the description of their work. I suggest to rewrite the introduction and to describe better the effects of the earthquake. Starting from

the avalanche it is important to define if it is an ice avalanche from glaciers or rock avalanche or another more complex phenomenon. a good definition of the effects of the earthquake is fundamental to give to lectors the possibility to evaluate the effectiveness of the approach proposed by the authors. Page 2 from line 7: the introduction describe what the authors want to describe in the manuscript, I'm not sure that the authors really satisfy this objectives. For this reason, I strongly suggest the authors to check the text and control that they describe all this topics. Page 2 chapter 2.1: the description of the study area is very short and poor. I suggest that the authors consider the possibility to improve both the geological and geomorphological aspect of the study area Page 4 chapter 3: this is the most important part of the paper, but it is also very hard to understand. Since it was not presented in the introduction a good description of what occurred in this area, now it is very critical for readers to understand what the authors have found. I suggest to rewrite this part of the article and to start the description from the evidence of the gravitational phenomena that caused the disaster and then to describe the effect in the lower part of the slope. One of the main limitation of this paper is that authors concentrate their description on the technical description of satellite images and results, but they did not pay too much attention to the description of the occurred events. I know that a correct reconstruction of the sequence of events is very hard, but I also think that if you want to present a methodology that use multiple remote sensing systems to describe the catastrophic collapse in Langtang Valley, at the end is mandatory have a description of the collapse and the sequence of events reconstructed by authors.

---

## Author Comment (AC1) · 24 Nov 2016

**Reply comments (AC1) for the interactive comments on "Multiple remote sensing assessment of the catastrophic collapse in Langtang Valley induced by the 2015 Gorkha Earthquake" by Hiroto Nagai et al.**

The authors thank the anonymous referee #1 for his/her valuable comments. We will improve the manuscript according to his/her comments as following:

General comments

This paper demonstrated an assessment of the sediments caused by a catastrophic avalanche, using Remote Sensing data, such as, ALOS-2, WorldView-3, ALOS World 3D, etc. The topic of this manuscript is quite interesting, because L-band (PALSAR-2) could penetrate the cloud and vegetation. In fact, catastrophic collapse (earthquake, debris flow, landslide, etc.) always seem to be associated with rain and vegetation. So, PALSAR-2 have a great potential to immediately indicate a catastrophic collapse and contribute to decision-making for such hazards in the monsoon season. However, this manuscript need more information to illustrate its conclusions. Below, I comment on the few things which I think can be improved.

We will improve our manuscript especially to clarify what is already known for this hazard, what remote-sensing techniques which we used can identify for the mountain hazard, and what we can mention from the technique for this specific hazard.

Specific comments (1) "Introduction", in this section, introduced too many information about study site (move it to the 2.1 section), but lack the background and innovation to this research, it can't attract the reader's interest immediately.

We remove "The Langtang Valley is one of…in the future. [P02L05-L09]". In terms of describing our motivation, we already know that was a catastrophic avalanche event including debris and glacier ice which completely destroyed a mountain village (Kargel et al., 2015; Fujita et al., 2016; Lacroix, 2016). Here we would like to emphasize what was happened there (further information than saying "avalanche") and what aspect can be identified using remote sensing techniques for such a catastrophic avalanche event. We will add here;

> "Damage detection by remote-sensing SAR technique has been applied for urban
> damaged area (e.g. Kobayashi et al, 2011; Yonezawa and Takeuchi, 2001; Tamura
> and El-Gharbawi, 2015; Watanabe et al., 2016), but almost no case for huge-scaled
> mountain hazard was done. Then we apply the SAR technique of damage detection for
> the avalanche case. In addition, detail interpretation of the damaged area by means of

*high-resolution optical satellite imagery coupled with sediment volume estimation*
*would suggest detail feature of this avalanche. In this study…"*
(2) "2.1 study site", I think you'd better add a location map of study site to help to understand
where is it.
We add a location map with satellite coverages before Fig. 1.
(3) "2.2 Synthetic aperture radar imagery", just defined normalized coherence decrease (NCD),
didn't explain what is Coherence calculation and how to calculate it, in addition, you can't leave
out the process and method to noises filter, it's too brief in this part.
**<Coherence calculation and its normalization >**
We would like to add further information on the paragraph from P03L03 "Not only…":
- We performed coherence calculation using interferometric phase information of SAR
which was explained by Plank (2014) in detail.
- Coherence can be calculated from two SAR images observing an identical place twice
from same orbit and an incidence angle.
- Coherence means similarity in terms of phase and intensity information of receiving
microwave which is calculated for a pair of SAR images by

$$\gamma = \frac{E\langle c_1 c_2^* \rangle}{\sqrt{E\langle c_1 c_1^* \rangle E\langle c_2 c_2^* \rangle}}$$

where $c_1$ and $c_2$ are the corresponding complex valued pixels of the two images, $c^*$ is the
complex conjugate of c, and *E* indicates the expected value. Detail mathematical
procedure is described in Touzi et al. (1999) and López-Martínez and Pottier (2007).
- Great change of surface feature between two observations results in lower coherence
(lower similarity, in other words).
- Other noisy influences including vegetation growth can be reduced by calculating
normalized differences with a coherence calculated from pre-hazard two images. The
normalized coherence decrease is calculated as;

$$\gamma_{\text{diff}} = \frac{\gamma_{\text{pre}} - \gamma_{\text{int}}}{\gamma_{\text{pre}} + \gamma_{\text{int}}}$$

where $\gamma_{\text{pre}}$ is the coherence value between two images before the earthquake (October 4,
2014, and February 21, 2015) and $\gamma_{\text{int}}$ is the coherence value between those over the
earthquake (February 21 and May 2, 2015).
- When $\gamma_{\text{int}}$ is calculated for images over a hazard, higher-valued pixels of $\gamma_{\text{diff}}$ indicates the
reduction of the similarity which have high potential of hazard-induced deformation or
destruction.

- Several previous studies have applied this method using L-band SAR for damage
detection in urban area (e.g. Kobayashi et al, 2011; Yonezawa and Takeuchi, 2001;
Tamura and El-Gharbawi, 2015; Watanabe et al., 2016), but we could not find such a
study applied for mountain hazard.
- Throughout this study, we aim to emphasize possibility of normalized conference
difference using L-band SAR for damage detection in the mountain regions.

**\<Noise filtering\>**
We would like to add further information and a figure on the paragraph from P03L15
"Secondly, numerous…";
1. Radius of a window circle is set as 15 pixel.
2. A mean value of the pixels in a circle is calculated.
3. The mean value is put at the center pixel of the circle.
4. Moving the circle, every pixel on the output image is filled with the mean values in
the same way.

[Figure]

(4)" 2.4 Post-event optical imagery and DSM", the post-event DSM is very important to
calculate the sediments volume, this paper just said "was produced by NTT DATA as its
commercial service", obviously it's not enough, And "relative calibration/validation of this
DSM and the AW3D DSM was performed and summarized in a supplementary material", I
didn't find the supplementary material.
We understand. After that sentence we would like to add information of;
- The DSM is generated by stereo photogrammetric method using two WV-3 images
acquired on the same date (May 8, 2015).

- Stereo area collect mode (26.2 km swath, 112 km path) was selected.

- Two images of (1) forward looking with cross-track tilting to the west hand (i.e. average off nadir angle: 27 deg., average target azimuth: 245 deg. / scene id:

104001000BA62E00) and (2) backward looking with cross-track tilting to the west hand (i.e. average off nadir angle: 27 deg., average target azimuth: 319 deg. / scene id:

104001000B3B2300) were acquired.

- Spatial resolution after cross-track tilt was 0.38 m, coarsened from 0.31 m due to tilting.

- DSM generation flow (i.e. Stereo matching, RPC ortho-rectification, pixel resampling, and DSM data output) was operated by NTT DATA with their original software, where geo-referencing process was supported by WV-3 accurate orbit information without any in-situ ground control point and resampled pixel spacing is 2 m.

- Officially announced specification shows a vertical accuracy of 4 m and a horizontal accuracy of 5 m as root mean square errors.

- Our calibration with AW3D DSM in the study region shows a standard deviation error of 1.5 m (described in the supplement material).

The supplementary material is provided from the right column here (circled in red below).

[Figure]

**(5) Is it possible to do field survey to verify the results?**

Fujita et al. (2016) performed an in-situ survey. They estimated the total volume of the avalanche sediment as $6.81\times10^6$ m$^3$, which is 109% of what we estimated. We will add their information to the discussion chapter.

**(6) Improve the quality of the figures**

We will put higher resolution figures in the revised version.

**Additional references (for AC1 and AC2):**

Fujita, K., Inoue, H., Izumi, T., Yamaguchi, S., Sadakane, A., Sunako, S., Nishimura, K., Immerzeel, W. W., Shea, J. M., Kayashta, R. B., Sawagaki, T., Breashears, D. F., Yagi, H., and Sakai, A.: Anomalous winter snow amplified earthquake induced disaster of the 2015 Langtang avalanche in Nepal, Nat. Hazards Earth Syst. Sci. Discuss., doi:10.5194/nhess-2016-317, in review, 2016.

López-Martínez, C., & Pottier, E. (2007). Coherence estimation in synthetic aperture radar data based on speckle noise modeling. Applied optics, 46(4), 544-558.

Plank, S. (2014). Rapid damage assessment by means of multi-temporal SAR—A comprehensive review and outlook to Sentinel-1. Remote Sensing, 6(6), 4870-4906.

Touzi, R., Lopes, A., Bruniquel, J., & Vachon, P. W. (1999). Coherence estimation for SAR imagery. IEEE Transactions on Geoscience and Remote Sensing, 37(1), 135-149.

Shiraiwa, T., & Watanabe, T. (1991). Late Quaternary glacial fluctuations in the Langtang valley, Nepal Himalaya, reconstructed by relative dating methods. Arctic and Alpine Research, 404-416.

Shiraiwa, T. (1994). Glacial fluctuations and cryogenic environments in the Langtang Valley, Nepal Himalaya. Contributions from the Institute of Low Temperature Science. Series A, 38, 1-98.

Watanabe, M., Thapa, R. B., Ohsumi, T., Fujiwara, H., Yonezawa, C., Tomii, N., & Suzuki, S. (2016). Detection of damaged urban areas using interferometric SAR coherence change with PALSAR-2. Earth, Planets and Space, 68(1), 131.

---

## Author Comment (AC2) · 24 Nov 2016

**Reply comments (AC2) for the interactive comments on "Multiple remote sensing assessment of the catastrophic collapse in Langtang Valley induced by the 2015 Gorkha Earthquake" by Hiroto Nagai et al.**

The authors thank the anonymous referee #2 for his/her valuable comments. We will improve the manuscript according to his/her comments as following:

In this manuscript, the authors describe the use of different remote sensing approaches for the identification of the effects of the 2015 Gorka Earthquake. In my opinion, the topic is very interesting and suitable for this journal, but the manuscript could be considered ready for the publication only after major revisions. In the following some suggestions for the authors:

We will improve our manuscript especially to clarify what is already known for this hazard, what remote-sensing techniques which we used can identify for the mountain hazard, and what we can mention from the technique for this specific hazard.

Page 1 line 30: in the abstract the authors describe an avalanche and they introduce that the paper will be focused on it. After, in the introduction, they introduce the presence of avalanche, but also landslides and other gravitational processes. For the reader is not very easy to understand which what happened in this area and then to follow the authors in the description of their work. I suggest to rewrite the introduction and to describe better the effects of the earthquake. Starting from the avalanche it is important to define if it is an ice avalanche from glaciers or rock avalanche or another more complex phenomenon. A good definition of the effects of the earthquake is fundamental to give to lectors the possibility to evaluate the effectiveness of the approach proposed by the authors.

We are sorry for this complicated expression. Now most of the material is considered as an avalanche including numerous boulders (debris) and possibly involving glacier ice along the path.

Details:

At an early time, Kargel et al. (2015) defined this event as a landslide, but they also mentioned "co-seismic snow and ice avalanches and rockfalls" with an image of lower surface temperature observed by Landsat-8 thermal infrared sensor. Lacroix (2016) defined it as a debris avalanche composed mostly of ice and discussed its triggers around the mountain ridge above two glaciers. Fujita et al. (2016) saw sediment boulders on the surface including melting ice (following two pictures) and rapid surface lowering after the quake both by in-situ survey, suggesting that contained ice and snow were melting under the debris. Also

Fujita et al. (2016) concluded that extremely heavy snowfall before the quake increased its
volume, coupled with a weather station data.
Therefore we think this event should be defined as "a catastrophic avalanche event including
debris and glacier ice" in our introduction chapter. Our finding from interpretation of a
high-resolution WV-3 image suggests several layers of the sediment. We will discuss more
in detail what we can interpret from our remote-sensing data in the revised manuscript,
coupled with those previous studies.

[Figure]

A picture on the sediment surface taken by H. Watanabe a member of an in-situ survey
carried out for/by Fujita et al. (2016). [Date: Oct. 21, 2015]

[Figure]

A closed-up picture showing exposed and melting ice. [Date: Oct. 21, 2015]

Page 2 from line 7: The introduction describe what the authors want to describe in the manuscript, I'm not sure that the authors really satisfy this objectives. For this reason, I strongly suggest the authors to check the text and control that they describe all this topics.

We remove "The Langtang Valley is one of…in the future. [P02L05-L09]". In terms of describing our motivation, we already know that was a catastrophic avalanche event including debris and glacier ice which completely destroyed a mountain village (Kargel et al., 2015; Fujita et al., 2016; Lacroix, 2016). Here we would like to emphasize what was happened there (further information than just saying "avalanche") and what aspect can be identified using remote sensing techniques for such a catastrophic avalanche event. We will add here;

> *"Damage detection by remote-sensing SAR technique has been applied for urban damaged area (e.g. Kobayashi et al, 2011; Yonezawa and Takeuchi, 2001; Tamura and El-Gharbawi, 2015; Watanabe et al., 2016), but almost no case for huge-scaled mountain hazard was done. Then we apply the SAR technique of damage detection for the avalanche case. In addition, detail interpretation of the damaged area by means of high-resolution optical satellite imagery coupled with sediment volume estimation would suggest detail feature of this avalanche. In this study…"*

Page 2 chapter 2.1: the description of the study area is very short and poor. I suggest that the authors consider the possibility to improve both the geological and geomorphological aspect of the study area.

We will add geological and geomorphological information written by Shiraiwa and Watanabe (1991) and Shiraiwa (1994).

Page 4 chapter 3: this is the most important part of the paper, but it is also very hard to understand. Since it was not presented in the introduction a good description of what occurred in this area, now it is very critical for readers to understand what the authors have found. I suggest to rewrite this part of the article and to start the description from the evidence of the gravitational phenomena that caused the disaster and then to describe the effect in the lower part of the slope. One of the main limitation of this paper is that authors concentrate their description on the technical description of satellite images and results, but they did not pay too much attention to the description of the occurred events. I know that a correct reconstruction of the sequence of events is very hard, but I also think that if you want to present a methodology that use multiple remote sensing systems to describe the catastrophic collapse in Langtang Valley, at the end is mandatory have a description of the collapse and the sequence of events reconstructed
by authors.

> *What occurred in this area*

An avalanche including numerous boulders (debris) and possibly involving glacier ice
occurred.

> *Rewrite this part of the article and to start the description from the evidence of the*
*gravitational phenomena*

We think that already-known information, not our work, should be written in the
introduction or involved in the discussion chapter. Therefore we will clarify more
specifically what was happened (existing knowledge) in the introduction and discuss it
with our findings in the discussion mentioning more essential aspects.

> *They did not pay too much attention to the description of the occurred events*

We understand that. We will improve that in the above-mentioned way.

**Additional references (for AC1 and AC2):**

Fujita, K., Inoue, H., Izumi, T., Yamaguchi, S., Sadakane, A., Sunako, S., Nishimura, K., Immerzeel, W. W., Shea, J. M., Kayashta, R. B., Sawagaki, T., Breashears, D. F., Yagi, H., and Sakai, A.: Anomalous winter snow amplified earthquake induced disaster of the 2015 Langtang avalanche in Nepal, Nat. Hazards Earth Syst. Sci. Discuss., doi:10.5194/nhess-2016-317, in review, 2016.

López-Martínez, C., & Pottier, E. (2007). Coherence estimation in synthetic aperture radar data based on speckle noise modeling. Applied optics, 46(4), 544-558.

Plank, S. (2014). Rapid damage assessment by means of multi-temporal SAR—A comprehensive review and outlook to Sentinel-1. Remote Sensing, 6(6), 4870-4906.

Touzi, R., Lopes, A., Bruniquel, J., & Vachon, P. W. (1999). Coherence estimation for SAR imagery. IEEE Transactions on Geoscience and Remote Sensing, 37(1), 135-149.

Shiraiwa, T., & Watanabe, T. (1991). Late Quaternary glacial fluctuations in the Langtang valley, Nepal Himalaya, reconstructed by relative dating methods. Arctic and Alpine Research, 404-416.

Shiraiwa, T. (1994). Glacial fluctuations and cryogenic environments in the Langtang Valley, Nepal Himalaya. Contributions from the Institute of Low Temperature Science. Series A, 38, 1-98.

Watanabe, M., Thapa, R. B., Ohsumi, T., Fujiwara, H., Yonezawa, C., Tomii, N., & Suzuki, S. (2016). Detection of damaged urban areas using interferometric SAR coherence change with PALSAR-2. Earth, Planets and Space, 68(1), 131.

---

## Author Response (AR1)

**[Authors' Response for nhess-2016-285]**

**Editor Decision: Reconsider after major revisions (further review by Editor and Referees)**

**(26 Nov 2016) by Prof. Dr. Paolo Tarolli / Comments to the Author:**

Dear Authors, your paper has been revised by two reviewers. They raised several critical issues that need to be fixed before the publication. You provided a detailed feedback during the

NHESS open discussion. I think you should have a chance to propose a revised version of your work. My recommendation is to accept this paper after major changes.

In submitting your revised version, please provide a detailed list of the changes made to the text, and a detailed list of your responses to the reviewers' comments.

Please note that this editorial decision does not guarantee that your paper will be accepted for final publication in NHESS. A decision will be made when the revised version will be available, and will be evaluated with the help of the same, or further reviewers.

Best regards / Paolo Tarolli

The authors appreciate continuous handling of this manuscript on NHESS. The manuscript was revised as follows. Revised parts are colored in red in a new manuscript and their pages and lines are denoted in a bold font in this document.

**Reply comments (AC1) for the interactive comments on "Multiple remote sensing**
**assessment of the catastrophic collapse in Langtang Valley induced by the 2015 Gorkha**
**Earthquake" by Hiroto Nagai et al.**
The authors thank the anonymous referee #1 for his/her valuable comments. We improved the
manuscript according to his/her comments as following:
General comments
This paper demonstrated an assessment of the sediments caused by a catastrophic avalanche,
using Remote Sensing data, such as, ALOS-2, WorldView-3, ALOS World 3D, etc. The topic
of this manuscript is quite interesting, because L-band (PALSAR-2) could penetrate the cloud
and vegetation. In fact, catastrophic collapse (earthquake, debris flow, landslide, etc.) always
seem to be associated with rain and vegetation. So, PALSAR-2 have a great potential to
immediately indicate a catastrophic collapse and contribute to decision-making for such hazards
in the monsoon season. However, this manuscript need more information to illustrate its
conclusions. Below, I comment on the few things which I think can be improved.
We improved our manuscript especially to clarify what was already known for this hazard,
what remote-sensing techniques which we used can identify the mountain hazard, and what
we can mention from the technique for this specific hazard.
Specific comments
(1) "Introduction", in this section, introduced too many information about study site (move it to
the 2.1 section), but lack the background and innovation to this research, it can't attract the
reader's interest immediately.
We moved "The Langtang Valley is one of…**[previous: P02L05-L09]**" to the end of the
section 2.1. **[new: P02L30]**. In terms of describing our motivation, we already know that was
a catastrophic avalanche event including debris and glacier ice which completely destroyed a
mountain village (Kargel et al., 2015; Fujita et al., 2016; Lacroix, 2016). Here we aim to
emphasize detail information (further than saying "avalanche") and what aspect can be
identified using remote sensing techniques for such a catastrophic avalanche event. We added
here;
**[new: P02L07]** *"Damage detection through SAR technique has been applied for*
*urban damaged areas (e.g., Kobayashi et al, 2011; Yonezawa and Takeuchi, 2001;*
*Tamura and El-Gharbawi, 2015; Watanabe et al., 2016), but almost no case for a*
*large-scale mountain hazard was studied. We apply SAR damage detection for the*
*avalanche case. In addition, a detailed interpretation of the damaged area by means*

*of high-resolution optical satellite imagery coupled with sediment volume estimation*

*would provide detailed features of this avalanche. In this study…"*

(2) "2.1 study site", I think you'd better add a location map of study site to help to understand where is it.

We added a location map with satellite coverage as **Fig. 1**.

(3) "2.2 Synthetic aperture radar imagery", just defined normalized coherence decrease (NCD), didn't explain what is Coherence calculation and how to calculate it, in addition, you can't leave out the process and method to noises filter, it's too brief in this part.

**<Coherence calculation and its normalization >**

We added further information on the paragraph from **P03L12** "Not only…":

*Not only the amplitude imagery but also the phase information emitted and received*

*by the synthetic aperture radar (SAR) contributes to the situational awareness. We*

*performed coherence calculation using interferometric phase information of SAR,*

*which was explained by Plank (2014) in detail. Coherence can be calculated from two*

*SAR images observing an identical place twice from the same orbit and incidence*

*angle, thereby achieving similar phase and intensity information of the receiving*

*microwave, which is calculated for a pair of SAR images by*

$$\gamma = \frac{E\langle c_1 c_2^* \rangle}{\sqrt{E\langle c_1 c_1^* \rangle E\langle c_2 c_2^* \rangle}} \qquad\qquad (1)$$

*where $c_1$ and $c_2$ are the corresponding complex-valued pixels of the two images, $c^*$ is*

*the complex conjugate of c, and E indicates the expected value. The detailed*

*mathematical procedure is described in Touzi et al. (1999) and López-Martínez and*

*Pottier (2007). A significant change in surface feature between two observations*

*results in lower coherence (in other words, lower similarity). Other noisy influences,*

*including vegetation growth, can be reduced by calculating normalized differences*

*with a coherence calculated from two pre-hazard images. The normalized coherence*

*decrease (NCD) is calculated as*

$$\gamma_{diff} = \frac{\gamma_{pre} - \gamma_{int}}{\gamma_{pre} + \gamma_{int}} \qquad\qquad (2)$$

*where $\gamma_{pre}$ is the coherence value between two images before the earthquake (October*

*4, 2014 and February 21, 2015), and $\gamma_{int}$ is the coherence value between the two*

*images over the earthquake (February 21 and May 2, 2015). These data were*

*acquired from a same orbit with a spatial resolution of 10 m. When $\gamma_{diff}$ is calculated*

| 88 | *for images over a hazard, higher-valued pixels of $\gamma_{diff}$ indicate the reduction of the* |
| 89 | *similarity, which has high potential of hazard-induced deformation or destruction.* |
| 90 | *Several previous studies applied this method using L-band SAR for damage detection* |
| 91 | *in urban areas (e.g., Kobayashi et al., 2011; Yonezawa and Takeuchi, 2001; Tamura* |
| 92 | *and El-Gharbawi, 2015; Watanabe et al., 2016), but no such study applied this* |
| 93 | *method for mountain hazard. Throughout this study, we aim to emphasize the* |
| 94 | *possibility of normalized conference difference by using L-band SAR for damage* |
| 95 | *detection in mountain regions.* |
| 96 | |
| 97 | **<Noise filtering>** |
| 98 | We added further information and a figure **(Fig. 2)** on the paragraph from **P04L03**; |
| 99 | *Numerous noises are removed by focal statistics. In the NCD raw image, all pixel* |
| 100 | *values are overwritten by the mean values within 15-pixel circles around each pixel* |
| 101 | *(Fig. 2). This filter emphasizes the concentration of high values, whereas the* |
| 102 | *homogeneously scattered high values are de-emphasized. The detailed steps are as* |
| 103 | *follows:* |
| 104 | *1.     The radius of a window circle is set as 15 pixels.* |
| 105 | *2.     A mean value of the pixels in a circle is calculated.* |
| 106 | *3.     The mean value is placed in the center pixel of the circle.* |
| 107 | *4.     Moving the circle, every pixel on the output image is filled with the mean* |
| 108 | *values in the same way.* |
| 109 | |

| 110 | (4)" 2.4 Post-event optical imagery and DSM", the post-event DSM is very important to |
| 111 | calculate the sediments volume, this paper just said "was produced by NTT DATA as its |
| 112 | commercial service", obviously it's not enough, And "relative calibration/validation of this |
| 113 | DSM and the AW3D DSM was performed and summarized in a supplementary material", I |
| 114 | didn't find the supplementary material. |
| 115 | We understand. After that sentence we added further information as; |
| 116 | **[P04L29]** *The DSM is generated by stereo photogrammetric method using two WV-3* |
| 117 | *images acquired on May 8, 2015 using stereo-area-collect mode (26.2 km swath, 112* |
| 118 | *km path). Two images that are (1) forward looking with cross-track tilting to the west* |
| 119 | *hand (i.e., average off-nadir angle: 27°, average target azimuth: 245° /scene id:* |
| 120 | *104001000BA62E00) and (2) backward looking with cross-track tilting to the west hand* |
| 121 | *(i.e., average off-nadir angle: 27°, average target azimuth: 319° /scene id:* |
| 122 | *104001000B3B2300) were acquired. Spatial resolution after cross-track tilt was 0.38 m,* |
| 123 | *coarsened from 0.31 m because of tilting. DSM generation flow (i.e., stereo matching,* |

*RPC ortho-rectification, pixel resampling, and DSM data output) was performed by*
*NTT DATA with their original software, where the geo-referencing process was*
*supported by WV-3 accurate orbit information without any in-situ ground control point*
*and a resampled pixel spacing of 2 m. Officially announced specification shows a*
*vertical accuracy of 4 m and a horizontal accuracy of 5 m as root mean square errors.*
*In two sites that are neighboring the sediment surface, relative calibration/validation of*
*this DSM and the AW3D DSM was performed and summarized in a supplementary*
*material, in which a standard deviation error of 1.5 m between WV-3 and AW3D DSM*
*is reported. A pan-sharpened image (high-resolution and composite-color image)*
*generated from one scene of the pair was orthorectified by an author with 178 tie points*
*onto the PRISM image taken on October 12, 2008.*
Acknowledgement contains new mention for cooperation by NTT DATA **[P10L16]**.
The supplementary material is provided from the right column here (circled in red below).

[Figure]

(5) Is it possible to do field survey to verify the results?
Fujita et al. (2016) performed an in-situ survey. They estimated the total volume of the
avalanche sediment as $6.81 \times 10^6$ m$^3$, which is 109% of what we estimated. We added their
information to the discussion chapter;
**[P09L21]** *Furthermore, Fujita et al. (2016) performed an in-situ survey from which*
*they estimated the total volume of the avalanche sediment as $6.81 \times 10^6$ m$^3$, which is*
*109% of what we estimated. Thus, a comparison with the satellite-based studies by*

*Kargel et al. (2015) and Lacroix (2016) indicates that our estimated sediment volume is*

*within the most equivalent order to that from the in-situ measurement by Fujita et al.*

*(2016).*

(6) Improve the quality of the figures

We have higher resolution figures in the revised version.

**Reply comments (AC2) for the interactive comments on "Multiple remote sensing assessment of the catastrophic collapse in Langtang Valley induced by the 2015 Gorkha Earthquake" by Hiroto Nagai et al.**

The authors thank the anonymous referee #2 for his/her valuable comments. We improved the manuscript according to his/her comments as following:

In this manuscript, the authors describe the use of different remote sensing approaches for the identification of the effects of the 2015 Gorka Earthquake. In my opinion, the topic is very interesting and suitable for this journal, but the manuscript could be considered ready for the publication only after major revisions. In the following some suggestions for the authors:

> We improved our manuscript especially to clarify what was already known for this hazard, what remote-sensing techniques which we used can identify the mountain hazard, and what we can mention from the technique for this specific hazard.

Page 1 line 30: in the abstract the authors describe an avalanche and they introduce that the paper will be focused on it. After, in the introduction, they introduce the presence of avalanche, but also landslides and other gravitational processes. For the reader is not very easy to understand which what happened in this area and then to follow the authors in the description of their work. I suggest to rewrite the introduction and to describe better the effects of the earthquake. Starting from the avalanche it is important to define if it is an ice avalanche from glaciers or rock avalanche or another more complex phenomenon. A good definition of the effects of the earthquake is fundamental to give to lectors the possibility to evaluate the effectiveness of the approach proposed by the authors.

> We are sorry for this complicated expression. Now most of the material is considered as an avalanche including numerous boulders (debris) and possibly involving glacier ice along the path. To review this proceeding, further information and a figure **(Fig. 5)** was attached at the beginning of section 4.2. as;
>
> > **[P08L27]** *At an early time, Kargel et al. (2015) defined this event as a landslide, but they also mentioned "co-seismic snow and ice avalanches and rockfalls" with an image of lower surface temperature observed by Landsat-8 thermal infrared sensor. Lacroix (2016) defined it as a debris avalanche composed mostly of ice and discussed its triggers around the mountain ridge above two glaciers. Fujita et al. (2016) confirmed sediment boulders on the surface, including melting ice (Fig. 7) and rapid surface lowering after the quake, through an in-situ survey, thereby suggesting that contained ice and snow were melting under the debris. Fujita et al. (2016) concluded*

*that extremely heavy snowfall before the quake increased its volume, a finding that was*
*coupled with weather station data. Therefore, we think this event should be defined as*
*"a catastrophic avalanche event including debris and glacier ice" in our introduction.*
*Our finding from the interpretation of a high-resolution WV-3 image suggests several*
*layers of the sediment. Multiple segments of the collapsed sediment classified with a*
*WV-3 image imply different sediment sources that have fallen continuously in a short*
*period of time, generating sediment layers (Fig. 5). We could…*

Also a new sentence was added to the abstract as;
**[P01L19]** *Our findings suggest that the avalanche event did not supply a*
*homogeneous snow-and-ice material with debris but supplied multiple kinds of*
*sediments from sequential collapse in a short period.*

Page 2 from line 7: The introduction describe what the authors want to describe in the
manuscript, I'm not sure that the authors really satisfy this objectives. For this reason, I strongly
suggest the authors to check the text and control that they describe all this topics.
We moved "The Langtang Valley is one of…**[previous: P02L05-L09]**" to the end of the
section 2.1. **[new: P02L30]**. In terms of describing our motivation, we already know that
was a catastrophic avalanche event including debris and glacier ice which completely
destroyed a mountain village (Kargel et al., 2015; Fujita et al., 2016; Lacroix, 2016). Here
we aim to emphasize detail information (further than saying "avalanche") and what aspect
can be identified using remote sensing techniques for such a catastrophic avalanche event.
We added here;
**[new: P02L07]** *"Damage detection through SAR technique has been applied for*
*urban damaged areas (e.g., Kobayashi et al, 2011; Yonezawa and Takeuchi, 2001;*
*Tamura and El-Gharbawi, 2015; Watanabe et al., 2016), but almost no case for a*
*large-scale mountain hazard was studied. We apply SAR damage detection for the*
*avalanche case. In addition, a detailed interpretation of the damaged area by means*
*of high-resolution optical satellite imagery coupled with sediment volume estimation*
*would provide detailed features of this avalanche. In this study…"*

Page 2 chapter 2.1: the description of the study area is very short and poor. I suggest that the
authors consider the possibility to improve both the geological and geomorphological aspect of
the study area.
We added geological and geomorphological information as;

**[P02L24]** *The Lantang valley consists of the Gosainkund gneiss zone (various*
*gneisses and granitic migmatite) and the Langtang Himal migmatite zone*
*(medium-grained garnet-mica-gneiss of granitic composition and coarse-grained*
*augen-gneiss) (Arita et al. 1973; Shiraiwa and Watanabe 1991). Six successive glacial*
*stages were recognized from an in-situ dating survey on moraine compositions*
*(Shiraiwa and Watanabe 1991; Shiraiwa, 1994). Relatively extensive glaciation in the*
*Langtang Stage (3650–3000 yr BP) is suggested in the late Quaternary. Permafrost is*
*not highly expected in this valley because of the large amount of winter snow, which*
*prevents deep freezing in winter (Shiraiwa, 1994).*

Page 4 chapter 3: this is the most important part of the paper, but it is also very hard to
understand. Since it was not presented in the introduction a good description of what occurred
in this area, now it is very critical for readers to understand what the authors have found. I
suggest to rewrite this part of the article and to start the description from the evidence of the
gravitational phenomena that caused the disaster and then to describe the effect in the lower part
of the slope. One of the main limitation of this paper is that authors concentrate their description
on the technical description of satellite images and results, but they did not pay too much
attention to the description of the occurred events. I know that a correct reconstruction of the
sequence of events is very hard, but I also think that if you want to present a methodology that
use multiple remote sensing systems to describe the catastrophic collapse in Langtang Valley, at
the end is mandatory have a description of the collapse and the sequence of events reconstructed
by authors.

An avalanche including numerous boulders (debris) and possibly involving glacier ice
occurred. Overview of this event has already been summarized in the introduction chapter
from **[P02L01]**. In addition, already known findings are reviewed at **[P08L27]** as noted
above. The results chapter is constructed by what we additionally found from satellite
observations highlighting technical topics, and instead we renamed chapter 4.2. as "Details
of the avalanche event" to integrate what was already known and what we found, aiming
new insight.

**Additional references (for AC1 and AC2):**

Fujita, K., Inoue, H., Izumi, T., Yamaguchi, S., Sadakane, A., Sunako, S., Nishimura, K.,
Immerzeel, W. W., Shea, J. M., Kayashta, R. B., Sawagaki, T., Breashears, D. F., Yagi, H.,
and Sakai, A.: Anomalous winter snow amplified earthquake induced disaster of the 2015
Langtang avalanche in Nepal, Nat. Hazards Earth Syst. Sci. Discuss.,
doi:10.5194/nhess-2016-317, in review, 2016.

López-Martínez, C., & Pottier, E. (2007). Coherence estimation in synthetic aperture radar
data based on speckle noise modeling. Applied optics, 46(4), 544-558.

Plank, S. (2014). Rapid damage assessment by means of multi-temporal SAR—A
comprehensive review and outlook to Sentinel-1. Remote Sensing, 6(6), 4870-4906.

Touzi, R., Lopes, A., Bruniquel, J., & Vachon, P. W. (1999). Coherence estimation for SAR
imagery. IEEE Transactions on Geoscience and Remote Sensing, 37(1), 135-149.

Shiraiwa, T., & Watanabe, T. (1991). Late Quaternary glacial fluctuations in the Langtang
valley, Nepal Himalaya, reconstructed by relative dating methods. Arctic and Alpine
Research, 404-416.

Shiraiwa, T. (1994). Glacial fluctuations and cryogenic environments in the Langtang Valley,
Nepal Himalaya. Contributions from the Institute of Low Temperature Science. Series A,
38, 1-98.

Watanabe, M., Thapa, R. B., Ohsumi, T., Fujiwara, H., Yonezawa, C., Tomii, N., & Suzuki, S.
(2016). Detection of damaged urban areas using interferometric SAR coherence change
with PALSAR-2. Earth, Planets and Space, 68(1), 131.

---

## Author Response (AR2)

**Reply to anonymous reviewers #1 and #2**

We appreciate your valuable comments to improve this study. We hereby revised our manuscript as described below. Newly added description in the manuscript are colored in red.

Reply to Reviewer #2

The paper describes well the management and processing of remote sensing data, but it is still poor from the geomorphological point of view. In particular, I suggest to describe better the event occurred starting from the abstract and then in the text.

For the abstract I suggest to describe better that (if I understood correctly) the work described the effect of the catastrophic avalanche deposit area. The paper analyzed the effects of different deposition processes that are the related to the catastrophic avalanche made by rock, ice and snow.

We massively updated the manuscript including detail interpretation of deposition sequence from the sediment layers identified with WV-3 high-resolution image. The abstract is revised as follows.

> … In the WV-3 image, surface features were classified into 10 groups. Our analysis suggests that the avalanche event contains a sequence of (1) fast splashing body with air blast, (2) muddy huge mass flowing, (3) less mass flowing from another source, (4) smaller amount of splashing and flowing mass, and (5) splashing mass without flowing at the east and west sides. By means of satellite-derived pre- and post-event digital surface models, differences in the surface altitudes of the collapse events estimated the total volume of the sediments as $5.51 \pm 0.09 \times 10^6$ m$^3$, most mass of which are distributed along the river floor and a tributary water stream. These findings contributes for detail numerical simulation of the avalanche sequences as well as source identification, and furthermore, altitude measurements after ice/snow melting would reveal a contained volume of melting ice and snow.

A good geomorphological description of the avalanche deposition area is missing in chapter 2 and a partial description of results from previous study, that considered also field surveys, are presented only at the end of the manuscript (chapter 4.2). I think that readers need these details at the beginning of the text, to understand better what is occurred and what has been already published on this catastrophic event. At the end of the paper, in the discussion, authors can consider again bibliography to compare obtained results.

Previously known facts are summarized in the method chapter as follows.

> ### 2.2 Avalanche event
>
> In this catastrophic event, co-seismic snow-and-ice avalanches and rockfalls with concurrent air blasts (Cadwalladr 2015). This contains multiple phenomena as described as "disaster-within-a-disaster" (Kargel, 2015). The sediment deposition is consists mostly of

accumulated snow and less dominantly of glacier ice (Fujita et al. 2016). Satellite-based thermal infrared observation on 5 days after the quake denoted the deposition has 10-20 K lower surface temperature than surrounding terrains (Kargel et al. 2016). Water stream of the Langtang river was blocked once by the deposition but quickly recovered as the ice-and-snow deposition was melted (Kargel et al. 2016). The materials near the river bed had less boulder and sand-rich deposition, suggesting that they are originated from snow avalanche (Fujita et al. 2016). From the sediment volume and catchment area on the mountain hill, original snow depth before the avalanche occurrence was estimated at 1.82 m in the catchment hillslopes (Fujita et al. 2016). A meteorological observation at a neighbouring glacier suggested four major snowfall events since Oct 2014 and an anomalous large amount of snow was charged before the quake. An interview reported that many hanging glaciers were cracked and huge pieces of ice falling occurred forming a cloud gathering snow and rocks with air blast (Cadwalladr 2015). However an in-situ survey suggested that detached glacier ice was less dominant than involved snow, represented by observed clear ice balls in the deposition (Fujita et al. 2016). After an following mas movement between 8 and 10 May, ice-and-snow melting decrease the sediment volume by 40% until Oct, 2015 (Fujita et al. 2016).

Multiple landslides was also reported (Cadwalladr 2015). Ice cliffs, exposure of ice-rich thick layer under a bolder-rich debris layer, are identified near the Langtang river, suggesting different timing of avalanche and subsequent rockfalls (Fujita et al. 2016). In the opposite-side north-facing steep slopes, debris materials were found at 200-m higher places above the deposition bottom, which suggested that they travelled at 63 m s$^{-1}$ (Kargel et al. 2016). On the other hand, avalanche entraining sand and silt was reported as "black avalanches" (Fujita et al. 2016). Post-event photographs and satellite images suggested debris materials originated from rockfall and landslide were not dominant in the deposition (Fujita et al. 2016; Kargel et al. 2016).

The related articles all reported trees fallen down to uniformed directions at the opposite-side north-facing slope (Cadwalladr 2015; Fujita et al. 2016; ICIMOD 2015; Kargel et al. 2016). This was caused by catastrophic air blast reaching 332 km h$^{-1}$ travelled up to neighbouring villages of Singdum and Mundu (Kargel et al. 2016). Location change of a boulder over the event suggest that it received a blast exceeding 50 m s$^{-1}$ (Fujita et al. 2016). In terms of collapse trigger, three separated main sources were suggested around the mountain peaks at 7000 m a.s.l. by snow cover thinning (~20 m) between April 2014 and May 2015 (Lacroix 2016). Hanging glacier detachment was considered by another study (Fujita et al. 2016). As described above, furthermore, anomalous winter snow seemed to amplify the sediment mass (Fujita et al. 2016). Topographic comparisons over the event revealed that the total mass of the sediment deposition was $6.81 \pm 1.54 \times 10^6$ m$^3$ before the second mass movement caused in 8-10 May (Fujita et al. 2016) and $6.95 \times 10^6$ m$^3$ including

*the second mass deposition (Lacroix 2016)*

Another important element is figure 5a; this picture represents the area covered by avalanche deposits. This area has been divided in different sectors and (in my opinion) lectors should have the possibility to understand which choices did authors for dividing this area in sectors (morphological, sedimentological, thickness of deposits?). Again, the description from the geomorphological point of view is too poor to have a complete description of the event. This paper present a remote sensing application, but if authors want to work with natural disaster, they should provide a (complete) description of the case study also using elements and data already published.

We totally changed the delineation method more quantitatively and repetitively using un-supervised classification. The new method is described as follows.

**2.4 Post-event optical imagery and DSM**

*Post-event optical satellite imagery and DSM were used to recognize the damaged situation in detail. A DigitalGlobe's satellite, WorldView-3 (WV-3) observed the Langtang valley on May 8, 2015, with a panchromatic sensor of 0.31 m spatial resolution and a multispectral sensor of 1.24 m spatial resolution to generate a set of pan-sharpened stereo pair imagery (Fig. 3). First, Area of Interest (AOI) is defined as that includes all sediment depositions. The complicated sediment outlines are delineated from the WV-3 near-infrared band, which appears the best clear contrast between the sediment depositions and the surface terrain, by means of a segmentation function of Iterative Self Organizing (ISO) cluster classifier in ArcGIS (e.g. Ball and Hall, 1965; Richards and Richards) (Fig. 4a). Other multispectral band images (Red, Green, and Blue) and the panchromatic image are synthesized into a pan-sharpened image (i.e. color imagery with 0.3-m spatial resolution). Using this image, the sediment depositions are divided into several groups based on visible characteristics of colors (dark or light) and deposition features (splashing, muddy, and flowing) (Figs. 4b-4f). After all the steps these images and delineated polygon layers are orthorectified with 174 tie points onto the ALOS pan-sharpened image taken on October 12, 2008.*
*Using the set of…*

Then we describe how we have classified the layers in the result chapter with closed-up images as follows.

**3.3 Collapse mapping with a post-event optical imagery**

Visual identification and mapping of the sediment depositions from the very-high-resolution WV-3 image resulted in 0.88 km² covering which was classified into 10 groups (A-J) (Fig. 3; Table 1). The group (A) (area: 0.16 km²) is characterized by dark muddy bottom to splashing uphill parts (Fig. 4b) where numerous trees fallen to the splashing direction are identified

as previous studies reported (Fig. 4c) (e.g. Kargel et al. 2015). The group (B) (area: 0.25 km$^2$) begins from the headwall just under a glacier with relatively lighter colour than (A) (Figs. 4b; 4d). It flows to the river floor with curved streaks (Figs. 3; 4d; 4e). In the river flow, it shows more mud-like feature with visible wrinkles as group (C) (area: 0.13 km$^2$), accumulating to the downstream and slightly to the upstream, maintaining the same colour (Fig. 4b). The group (D) (area: 0.14 km$^2$) basically has clearly darker surface than (B) and (C) with less streaks and several splashed patches (Fig. 4e). Simultaneously gradual colour transition is seen from (D) to (B) (Fig. 4e). The group (E) (area: 0.02 km$^2$) is located at the lower side of (D) with the same colour and rather muddy feature quite like (C) (Fig. 4e). Gradual colour transition is also seen from (D) to (B). On the east side, very dark-colour patches of (F) (area: 0.02 km$^2$) and detached parts (G) (area: 0.01 km$^2$) are found (Figs. 3). They seem splashing, but have relatively muddy feature and not so homogeneous directivity compared to (A). Dark aperture deposition of (H) (area: 0.07 km$^2$) begins from another headwalls which is wider than and is independent from that for (B) (Fig. 4f). The splashing parts are blocked by (B) and (C), whereas the western part starts flowing along a narrow path to the river floor grouped as (I) (area: 0.02 km$^2$) (Fig. 3). This flow is finally connected to and covers (C) (Fig. 4c). The group (J) (area: 0.05 km$^2$) is a parallel and more aperture/splashing deposition compared to (H) (Fig. 4f). The surface colour varies from lighter to darker than (J), not related to the flow path.

Page 7 line 7: In this paragraph, authors described the volume change analysis. In particular, they assumed that negative changes are mistakes and they correct them to zero. In my experience, the transition zone of large landslides (and in particular of rock avalanches) is often characterized by negative changes that are caused by the erosional effect of the huge mass and its velocity. Of course I hadn't the possibility to analyze "uncorrected" data, but I suggest that authors consider the possibility that this complex process can cause also negative changes during its runout.

The interpretation of DEM comparison results is often a very complex task. The recognition of real negative changes from artefacts can be usually based on the shape and distribution of negative areas. A morphological validation can be usually considered a good solution.

As advised, we considered negative altitude changes using DoD tool recommended by another reviewer. Especially, the north-facing steep slope corresponding to the sediment group (A) seems to have lower reliability on DSM generation as entire negative trend. Destruction of forest and buildings as well as surface erosion are also considered in the revised manuscript as follows.

DoD3.0: http://gcd.joewheaton.org/downloads/older-versions/dod-3-0

**3.4 Surface elevation changes**

…Altitude decreasing was denoted in the groups (A), (F), and (G), where dominance of

surface erosion and DSM error are considered. Mean altitude changes in the groups (D), (H), and (J) are smaller than the defined uncertainty level, 1.5 m.

Calculating the altitude change and surface area, a total deposition volume of $5.51\pm0.09\times10^6$ m$^3$ was estimated, which is included within the estimated volume range by Fujita et al. (2016) ($6.81\pm1.54\times10^6$ m$^3$) and not larger than the volume including the second mass movement ($6.95\times10^6$ m$^3$) (Lacroix 2016). In addition, total eroded volume of $1.64\pm0.06\times10^6$ m$^3$ was estimated, most of which belongs to the group (A). In addition to the effect of the fallen trees, fundamental bias error induced by WV-3 DSM generation is considered for this extremely steep slope, because splashed patches and muddy deposition both denotes negative values. As well, groups (F) and (G) have negative net volume difference, possibly because of building collapse and slightly negative DSM bias larger than the deposition volume of the dark-colour materials.

Considering language, I suggest to revise in particular the following paragraphs because they are not very clear:
Page 1, line 12 – 17: please consider to rewrite this paragraph because it is not very clear

We revised that to follows.

> *The main shock of the 2015 Gorkha Earthquake in Nepal induced numerous avalanches, rockfalls, and landslides in Himalayan mountain regions. A major village in the Langtang valley was destroyed with numerous victims by a catastrophic avalanche event, which consists of snow, ice, rock, and blast wind. The hazard process is understood mainly depending on limited witness, interview, and an in-situ survey after a monsoon season. To record immediate situation and to understand deposition process, we performed an assessment by means of satellite-based observations carried out in no later than two weeks after the event. The avalanche-induced sediment deposition was delineated with calculation of decreasing coherence and visual interpretation of amplitude images acquired from the Phased Array-type L-band Synthetic Aperture Radar-2 (PALSAR-2). These outlines area highly consistent with that delineated from a high-resolution optical image of WorldView-3 (WV-3). The delineated sediment areas were estimated as 0.63 km2 (PALSAR-2 coherence calculation), 0.73 km2 (PALSAR-2 visual interpretation), and 0.88 km2 (WV-3), respectively. In the…*

Page 6, lines 14 and 15: sectors 1 to 4 are considered "dark" but also sectors 12 to 15 are considered dark too, is it correct?

The chapter 3.3 was completely rewritten as described above. The sediment deposition has several dark parts in separated locations. We distinguish them with differences of surface features (flowing or splashing, for example).

Page 7, line 7 to 15: please consider to rewrite this paragraph because it is not very clear

We changed the method and results, adding consideration of negative altitude changes. The description on this line was therefore rewritten as follows.

> *Calculating the altitude change and surface area, a total deposition volume of $5.51\pm$ $0.09\times10^6$ m$^3$ was estimated, which is included within the estimated volume range by Fujita et al. (2016) ($6.81\pm1.54\times10^6$ m$^3$) and not larger than the volume including the second mass movement ($6.95\times10^6$ m$^3$) (Lacroix 2016). In addition, total eroded volume of $1.64\pm0.06\times10^6$ m$^3$ was estimated, most of which belongs to the group (A). In addition to the effect of the fallen trees, fundamental bias error induced by WV-3 DSM generation is considered for this extremely steep slope, because splashed patches and muddy deposition both denotes negative values. As well, groups (F) and (G) have negative net volume difference, possibly because of building collapse and slightly negative DSM bias larger than the deposition volume of the dark-colour materials.*

**(1) You can use DEM of difference (DoD) algorithm to estimate the Surface elevation changes.**

As advised, we considered negative altitude changes using DoD tool recommended here. Summary is described in Table 1. Especially, the north-facing steep slope corresponding to the sediment group (A) seems to have lower reliability on DSM generation as entire negative trend. Destruction of forest and buildings as well as surface erosion are also considered in the revised manuscript as follows.

DoD3.0: http://gcd.joewheaton.org/downloads/older-versions/dod-3-0

**(2) You should refine the "discussion", not only present the result (you already write in the part of "result", you should discuss the "advantages, disadvantages, future work et al.,")**

One core part of discussion was replaced as follows. we demonstrated temporal sequence of sediment deposition layers from identified layer orders. Connection to future studies is also describes at the end.

**4.3 Temporal sequence of the avalanche event**

[revised manuscript text omitted]